# How Do LLMs Use Their Depth?

## Abstract

Growing evidence suggests that large language models do not use their depth uniformly, yet we still lack a fine-grained understanding of their layer-wise prediction dynamics. In this paper, we trace the intermediate representations of several open-weight models during inference and reveal a structured and nuanced use of depth. Specifically, we propose a "*Guess-then-Refine*" framework that explains how LLMs internally structure their computations to make predictions. We first show that the top-ranked predictions in early LLM layers are composed primarily of high-frequency tokens, which act as statistical guesses proposed by the model early on due to the lack of appropriate contextual information. As contextual information develops deeper into the model, these initial guesses get refined into contextually appropriate tokens. Even high-frequency token predictions from early layers get refined $> 70\%$ of the time, indicating that correct token prediction is not "one-and-done". We then go beyond frequency-based prediction to examine the dynamic usage of layer depth across three case studies. (i) Part-of-speech analysis shows that function words are, on average, the earliest to be predicted correctly. (ii) Fact recall task analysis shows that, in a multi-token answer, the first token requires more computational depth than the rest. (iii) Multiple-choice task analysis shows that the model identifies the format of the response within the first half of the layers, but finalizes its response only toward the end. Together, our results provide a detailed view of depth usage in LLMs, shedding light on the layer-by-layer computations that underlie successful predictions and providing insights for future works to improve computational efficiency in transformer-based models.

## 1 Introduction

Despite the remarkable performance of large language models (LLMs), their internal computations remain poorly understood. One critical question is: how do LLMs internally structure their computations during inference and use their depth layer-by-layer to arrive at predictions? Are specific token predictions always computed at the last layer or does the model settle on predictable tokens early on and simply propagate these predictions? These questions have implications both for interpreting the internal computations of these models and for building more efficient LLM that can use their compute dynamically.

The LogitLens framework (Nostalgebraist, 2020) provides some initial insights by using the final unembedding layer of a model to decode intermediate layer representations in the token space. This approach shows that, sometimes, generated tokens already begin to appear as top-ranked predictions at intermediate layers, sometimes long before the final layer. Another tool, TunedLens (Belrose et al., 2023), adapts the original LogitLens approach by learning a linear transformation between a model's intermediate layer and the final layer, which enables tracking of token-level predictions without assuming that they lie in the same space. Here, we leverage TunedLens to detail and quantify token prediction patterns across LLM layers.

We propose a "*Guess-then-Refine*" framework for understanding model computations across layers, where models first create early guesses that get refined in subsequent layers. We first show that early-layer predictions in LLMs are largely composed of high-frequency tokens. For example, for Pythia6.9B, while the Top-10 most frequent tokens account for about 34% of the final predictions, the top-1 ranked early layers predictions contain the Top-10 tokens more than 75% times. We call this "*Frequency-Conditioned Onset*", where LLMs tend to use corpus statistics while making

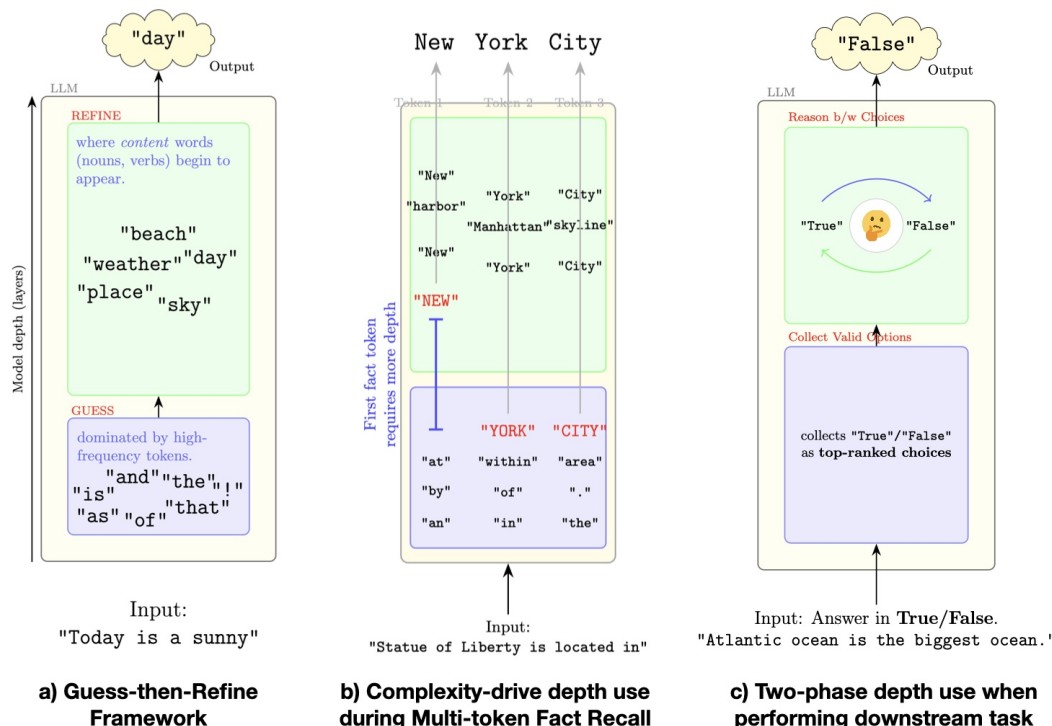

a) Guess-then-Refine Framework

b) Complexity-drive depth use during Multi-token Fact Recall

c) Two-phase depth use when performing downstream task

Figure 1: Illustrative depictions of our *Guess-then-Refine* framework and complexity-aware depth use in LLMs. **(a)** Early layers elevate high-frequency tokens as statistical guesses, which deeper layers refine into context-appropriate predictions (*Frequency-Conditioned Onset*). **(b)** Multi-token fact recall: the first token of the answer requires the largest depth on average, while subsequent tokens appear at much shallower depths. **(c)** Option-constrained downstream tasks (e.g., True/False): early layers collect valid options into the top ranks, while later layers reason between top choices to generate final answer. **Overall:** LLMs act as *early statistical guessers and late contextual integrators*, exhibiting *complexity-driven depth use*.

candidate proposals in early layers. We then show that these early layer proposals are infact guesses, which get heavily refined in subsequent layers, where more than 80% of the early layer predictions get refined into into contextually appropriate generations by the final layer.

We also uncover that LLMs use their depth dynamically based on the task at hand. (i) First, we show that during regular next token prediction, easier to predict tokens corresponding to punctuation and function words begin to get predicted correctly by early-to-mid LLM layers, whereas harder to predict tokens like nouns and verbs appear much later. (ii) We then look at multi-token fact recall task, where the model is asked factual questions whose answers span multiple tokens. We see that the second and third answer token predictions appear much earlier than the first, indicating a higher computational load associated with selecting the initial response direction. (iii) Finally, we analyze the layer-wise prediction dynamics when the models perform downstream tasks with a constrained set of option choices, such as answering multiple-choice (MCQ) or True/False questions. We show this task can be broken down into two steps: by the middle layers, the model arrives at the valid option choices as top-ranked predictions, and at the later layers, it deliberates between the valid options to reach the final answer. These experiments are indicative of *Complexity-Driven Depth Use*, where easier tasks require fewer layers of computation.

Thus, through various experiments, we show that LLMs internally structure their computations following a "Guess-then-Refine" strategy, where they propose high-frequency tokens as top-ranked proposals in early layers, which later get modified into contextually appropriate tokens in deeper layers. We also show that LLMs are natural dynamic depth models and use their depth intelligently based on the complexity of the task. More complex predictions require larger depth, while eas-

ier computations finish earlier. We perform this analysis on four open-weight models - GPT2-XL (Radford et al., 2019), Pythia-6.9B (Biderman et al., 2023), Llama2-7B (Touvron et al., 2023) and Llama3-8B (Meta, 2024). This analysis is done using the TunedLens probe (Belrose et al., 2023) probe, which allows us to decode earlier layer representations with higher fidelity. We also perform ablations on TunedLens to confirm that our results represent information content in early layers.

## 2 BACKGROUND : TUNEDLENS

Intermediate activations in LLMs can be projected onto the vocabulary space using the unembedding matrix (Nostalgebraist, 2020). Let $h^l$ represent the hidden representation after layer $l$. LogitLens (Nostalgebraist, 2020) projects these representations in the token space according to the following equation:

$$\texttt{LogitLens}(h^l) = W_U \Big[ \texttt{Norm}_f \big[ h^l \big] \Big] \tag{1}$$

where $\texttt{Norm}_f$ represents the final normalization[1] applied before the application of the unembedding matrix $W_U \in \mathbb{R}^{d \times |V|}$; where $|V|$ represents the vocabulary size, and $d$ is the hidden dimension size of the model.

Prior work has shown that directly applying the unembedding matrix to intermediate layers can lead to unreliable results as intermediate features may operate in different subspaces compared to the final layer outputs (Din et al., 2023; Belrose et al., 2023). We therefore opt to use TunedLens for more robust results. TunedLens learns an affine mapping between the output representations of each layer and the final unembedding matrix in the form of *translators* ($A_l \in \mathbb{R}^{d \times d}, b_l \in \mathbb{R}^d$), as shown below:

$$\texttt{TunedLens}(h^l) = \texttt{LogitLens}(A_l h^l + b_l) \tag{2}$$

The TunedLens probes are trained to minimize the KL-divergence between the final layer probability distribution and an intermediate layer probability distribution according to the following equation:

$$\underset{A_\ell, b_\ell}{\arg\min} \, \mathbb{E} \big[ D_{KL} \big( f_\theta(h^l) \, \| \, \texttt{TunedLens}(h^l) \big) \big] \tag{3}$$

where $f_\theta(h)$ represents the probability distribution for input representation $h^l$ at the output of the model. Thus, TunedLens translators ($A_\ell, b_\ell$) are trained to allow for better transferability between intermediate layers and the final unembedding matrix, which provide more faithful decoding of intermediate representations of the model, especially in the early layers (Belrose et al., 2023). This is why we opt to use TunedLens for for decoding the intermediate representations of LLMs.

## 3 FREQUENCY-CONDITIONED ONSET IN EARLY LLM LAYERS

We begin by studying the top-ranked predictions across LLM layers during next token prediction using the TunedLens probe. We look at four open-weight models - GPT2-XL (Radford et al., 2019), Pythia-6.9B (Biderman et al., 2023), Llama2-7B (Touvron et al., 2023) and Llama3-8B (Meta, 2024). We use the TunedLens probes provided by the original authors for these models[2].

**Methodology:** We divide vocabulary tokens of each model into four buckets by their frequency of occurrence in a corpus. For this, we take the tokenizer of each model and tokenize the English Wikipedia as distributed via the Hugging Face Datasets library (Lhoest et al., 2021), and record the frequency of each token. We use the following four buckets - (i) Top1-10 bucket, which contains the top 10 most frequent tokens in a corpus and accounts for approximately 23-25% of the corpus depending on the model, (ii) Top11-100 bucket, which contains the next 90 most frequent tokens

---

[1] We use normalization to mean both LayerNorm Ba (2016) used in GPT-family of models and RMSNorm Zhang & Sennrich (2019) used in the Llama-family of models.

[2] https://huggingface.co/spaces/AlignmentResearch/tuned-lens/tree/main

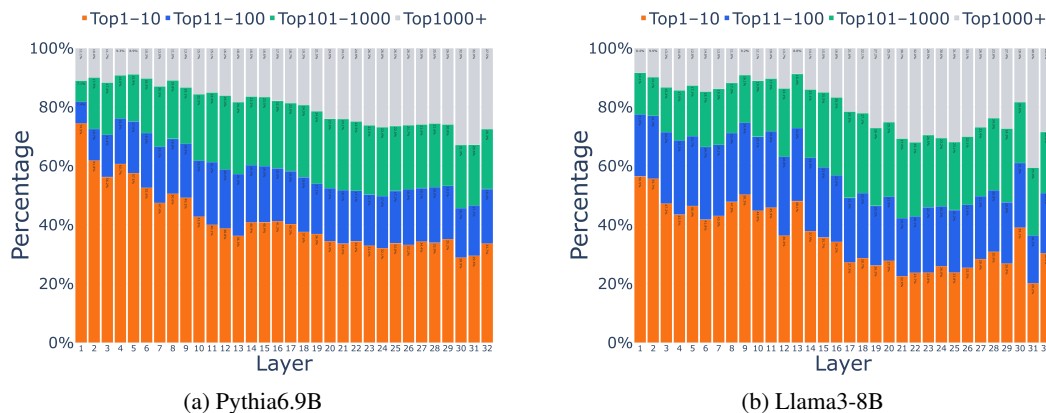

(a) Pythia6.9B          (b) Llama3-8B

Figure 2: **Frequent tokens dominate early-layer predictions.** At each layer, the layer's top-1 ranked token is grouped by corpus-frequency buckets (Top1–10, Top11–100, Top101–1000, Top1000+). Early layers are dominated by Top1–10 bucket tokens, where LLMs make statistical guesses and propose high-frequency tokens as top-ranked candidates under limited information, while deeper layers increasingly replace them with rarer, context-appropriate tokens.

and accounts for the next 16-20% of corpus tokens, (iii) `Top101-1000` bucket, which contains the next 900 most frequent tokens and accounts for 16-21% of corpus tokens, and finally the (iv) `Top1000+` bucket, which contains the remaining tokens and accounts for 35-40% of the corpus.

We evaluate the model during the next-token prediction task, where the base model is provided with prefixes extracted from the English Wikipedia corpus, cut-off at random points, and track the top-ranked token at each intermediate layer as the model predicts the next token. These prefixes have a variable lengths and can span multiple sentences (see Appendix A for details).

### 3.1 LLMs are Early Statistical Guessers

The top-ranked predictions at each intermediate layer classified into corresponding frequency buckets are shown in Figure 2. **We discover that early layer predictions are heavily dominated by tokens belonging to the `Top1-10` bucket**. For example, more than 75% top-ranked tokens for Pythia-6.9B (Figure 2a) belong to the `Top1-10` bucket at the first layer, while only 33% of the final predictions eventually made by the model lie in the `Top1-10` bucket. Similarly for Llama3-8B (Figure 2b), the early layer predictions belong to the `Top1-10` bucket for about 57% prefixes at the first layer, while the final predictions lie in the `Top1-10` bucket only about 30% of times. A similar trend is observed in GPT2-XL and Llama2-7B, as shown in Appendix B.

This observation shows that LLMs follow a very specific pattern when making initial proposals, which can be explained by understanding the amount of information models have to work with in early layers when compared to deeper layers. In early layers, the model has incomplete contextual information about an input sentence, since the input has not been through enough attention layers to aggregate a complete contextual representation of the input. Additionally, in early layers, the model is also unable to access the factual knowledge stored in its parameters to make a correct predictions, since the knowledge stored in model parameters usually exists within the middle MLP layers of a model (Geva et al., 2020; Meng et al., 2022; Geva et al., 2023). In the absence of enough contextual information and learnt knowledge, the best strategy is to revert back to corpus statistics. To explain this, let us take the example of an extreme scenario where we do not have access to any information about the input prefix. In such a situation, if one had to guess the next token, picking a token from the `Top1-10` bucket, which accounts for about 25% of the corpus, would be the best strategy as it maximizes the chances of being correct with a minimum set of tokens. Such an intuitive strategy naturally emerges out of the optimization pressures of the pre-training process.

**Thus, LLM make statistical guesses by promoting the most frequent tokens as top-ranked proposals in early layers.** As contextual information develops deeper into the model, these guesses

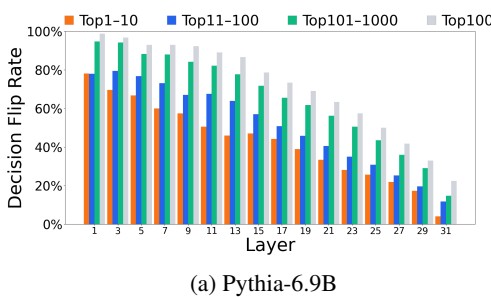 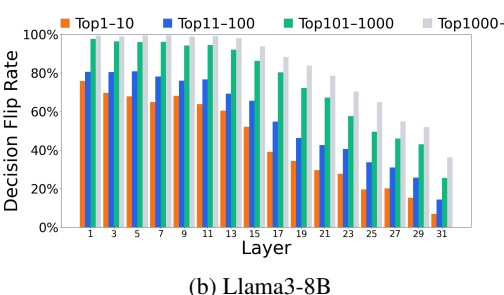

(a) Pythia-6.9B                   (b) Llama3-8B

Figure 3: **Early-layer predictions are heavily revised.** For each layer and frequency bucket, we show the fraction of top-1 ranked predictions that are overturned by the final layer ("decision flip rate"). We see that about 80% of layer-1 Top1–10 predictions change by the end, and this rises to ˜100% for Top101–1000 and Top1000+ bucket tokens. As expected, flip rates drop with depth.

based on corpus statistics get refined into appropriate predictions, which is when lower frequency tokens begin to appear.

## 3.2 MASSIVE CONTEXTUAL REFINEMENT IN LATER LAYERS

In the previous section, we show that early LLM layers propose candidates based on corpus frequency. In this section, we show that early layer predictions are in fact *guesses*, and as contextual information builds up as we go deeper into the model, these early suggestions get *heavily* refined into contextually appropriate tokens.

Indications for later layer refinements can already be seen in Figure 2, where the top-ranked tokens in early layers belong to the Top1000+ bucket for only about 10% of the prefixes, a number that increases by around 300% by the final layer. However, here we consider a more specific metric for refinement. We define *refinement* with the "flip" metric which measures the percentage of top-1 ranked tokens at an intermediate layer that are not the same as the final prediction. Hence, these intermediate layer predictions get refined into the correct prediction in deeper layers, which objectifies our refinement process. Figure 3 shows a clearer picture of the magnitude with which early layer proposals in LLMs get refined to final predictions. For Pythia-6.9B (Figure 3a), while early layer decisions are dominated by the Top1-10 bucket tokens as shown in previous section, almost 80% of those predictions get modified by the final layer. This is also true for Llama3-8B (Figure 3b). Additionally, if a prediction at layer 1 lies in the Top1000+ bucket, it gets modified by the final layer with almost 100% probability. Results for the remaining models can be found in Appendix B. **This shows that early layer proposals get modified in overwhelming numbers by the time they reach the final layer** and indicates that refinement happens for both high- and low-frequency tokens.

Thus, early LLM layers propose frequency-conditioned guesses that are highly non-permanent, where approximately 60–80% of early top-ranked guesses are eventually replaced. On the other hand, permanence rises with depth along with context integration.

## 4 COMPLEXITY-DRIVEN DEPTH USE IN LLMS

In the previous section, we saw that LLMs propose initial guesses in early layers based on corpus statistics, which get refined in large numbers as more contextual information develops through the model. In this section, we go into more details of this guess-then-refine process across layers. Specifically, we show that model depth in LLMs gets used flexibly, where easier to execute tasks (or subtasks) finish at shallower depths, while more complicated tasks are left for later layers. We show this using three case studies: next-token prediction, fact recall, and fixed-response tasks. We perform all experiments for GPT2-XL, Pythia-6.9B, Llama2-7B and Llama3-8B. We present the results for Pythia-6.9B and Llama3-8B in the main paper and results for the remaining models can be found in Appendix C.

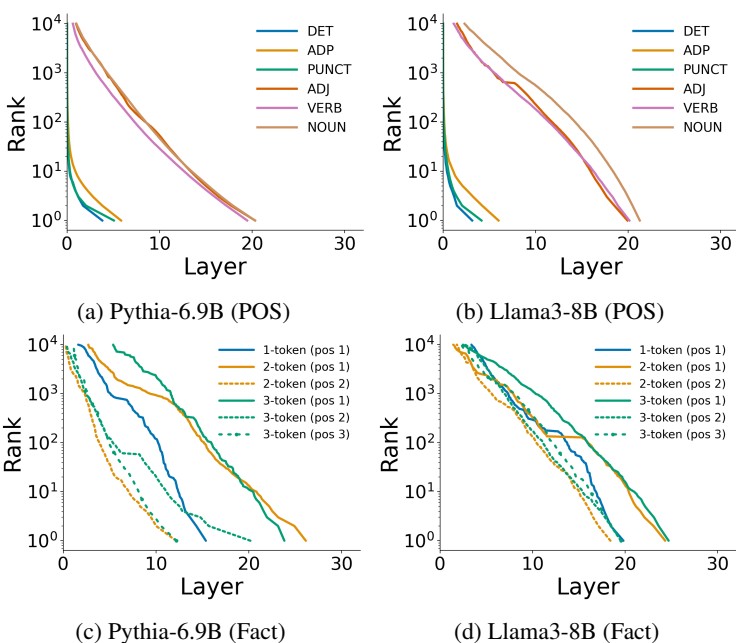

(a) Pythia-6.9B (POS)  (b) Llama3-8B (POS)

(c) Pythia-6.9B (Fact)  (d) Llama3-8B (Fact)

Figure 4: **Earliest crossing thresholds of predicted tokens by POS (top) and for multi-token facts (bottom).** We determine which token the model predicts at the final layer and then track the earliest layer (x axis) at which the TunedLens rank of that token crosses a given threshold (y axis). **(a, b)** Case Study I: When predicted tokens are grouped by POS category, tokens corresponding to function words (DET, ADP) and punctuation appear much earlier than content words (ADJ, VERB, NOUN). **(c, d)** Case Study II: We plot tokenwise results for fact recall responses that consist of 1 token, 2 tokens, or 3 tokens. E.g., 3-token facts consist of 3 tokens at positions (pos) 1, 2, and 3. In both models, the first tokens of multi-token facts appear much later than subsequent tokens.

## 4.1 CASE STUDY I: DEPTH USE BY POS CATEGORY

**Methodology:** We first study the depth use strategy in LLMs during next-token prediction. Here, instead of looking at the top-ranked prediction at each layer, *we track how the predicted token becomes top-ranked through the model depth*. To do this, we send a prefix text as input to a model, determine the next predicted token, and then track the rank of the predicted token through the intermediate layers. The predicted token is classified into one of the following part-of-speech (POS) categories - determiners (DET), adpositions (ADP), punctuations (PUNCT), adjectives (ADJ), verbs, and nouns. We use POS to categorize generated tokens because they give us an indication of the role a generated word plays in a sentence. POS categories like DET and ADP contain *function words* that do not provide much meaning to a sentence, whereas other categories like adjectives, verbs, and nouns are made up of *content words* where the main information content of a sentence is present. We plot the average layer at which a particular rank threshold is crossed by the predicted token at intermediate layers.

For example, if the input prefix sentence is *"Today is a sunny"* and the model predicts *"day"* as the generation, we first use Spacy to classify the POS of the predicted word. Then, we use TunedLens to trace the predicted rank of *"day"* at different layers within the model. The first layer inside a model at which the predicted token crosses a particular rank threshold is recorded[3].

**Results:** Figures 4a and 4b show the average number of layers required for a predicted token to first cross a rank-threshold. Function words (DET and ADP) and punctuation (PUNCT) progress and reach rank-1 much faster than content words (ADJ, VERB, NOUN). For example, when the predicted word is a DET, it first reaches rank 1 on average around layer 5 for both Pythia-6.9B and

---

[3]Rank progression in intermediate layers is non-monotonic, and often top-ranked tokens at an intermediate layers can then increase in rank in later layers. However, here we record the first evidence of the predicted token crossing a particular rank threshold to track the minimum computation required to generate a token.

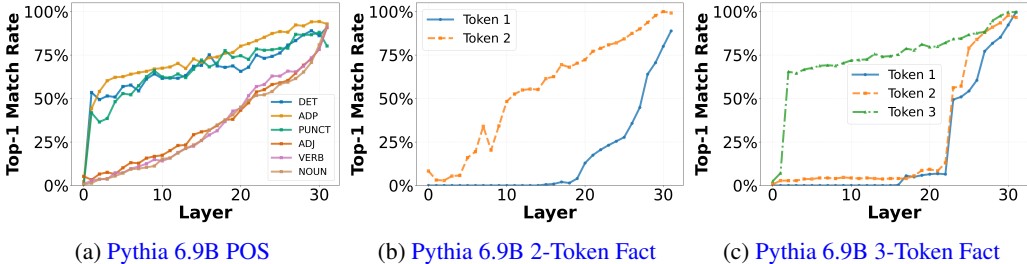

(a) Pythia 6.9B POS  (b) Pythia 6.9B 2-Token Fact  (c) Pythia 6.9B 3-Token Fact

Figure 5: **Early exiting experiments.** To test our hypothesis that easy to predict token categories finish processing earlier more often, we find the Top-1 match rate for an intermediate-layer decoded token with the final prediction - the match is successful if the Top-1 ranked prediction at a given layer matches the final prediction. The decoding is done via TunedLens. In (**a**), we see that the match rate in early layers is much larger for function tokens belonging to DET, ADP and PUNCT tokens, whereas it is much lower for content token categories of ADJ, VERB and NOUN. In (**b, c**), we see that once the first token is predicted, the subsequent tokens are easier to predict at earlier layers. The Top-1 match rate for the second token in 2-token facts, and the third token in 3-token facts is much higher in earlier layers. Similar results can be seen for other models in Figure 18.

Llama3-8B, while other content words categories reach rank-1 much deeper into a model, closer to layer 20. Post-stability analysis where we measure the average layer at which predicted token crosses a given rank threshold for the *final* time are presented in Figure 16 in Appendix. The qualitative results remain the same where the rank progression of DET, ADP and PUNCT tokens is much faster when compared to content tokens, however the phenomena is shifted deeper into the models as expected for post-stability calculations.

This result is also consistent with the finding in Section 3: DET, ADP, and PUNCT contain primarily high-frequency tokens and are therefore predicted early in the model, at the initial guessing stage. Content words, however, must rely on contextual inference in middle layers to be guessed correctly.

**Early-Decoding Experiments:** To validate that the information processing trajectories when predicting function tokens proceeds faster than content tokens, we perform early decoding experiments at intermediate layers using TunedLens and calculated the percentage of times the intermediate Top-1 rank prediction matches the final prediction at an intermediate layer. Figure 5a clearly shows that the Top-1 match rate is much higher for function tokens (DET, ADP) and PUNCT in early layers, whereas it is much lower for content token categories. To quantify this difference, at layer 10, it is almost twice as likely for the Top-1 prediction to be the final token when the final token belongs to the function token or PUNCT categories than when it belongs to content token categories. These results have direct implications for future work in early-exiting.

Overall, Case Study I suggests that easier-to-predict tokens are processed at a faster rate and get guessed earlier in the model more often than harder-to-predict tokens.

## 4.2 CASE STUDY II: DEPTH USE DURING MULTI-TOKEN FACT RECALL

**Methodology:** In this case study, we track the predicted tokens through intermediate layers during fact recall using the MQuAKE dataset (Zhong et al., 2023), which contains a variety of single-token and multi-token facts. An example query that is input to the model in this dataset is - "The Statue of Liberty is located in" and the expected generated token is "New York City". We only study internal computations in scenarios where the model predicts the correct answer. We also split the dataset into three different cases where the answer contains one, two or three tokens and look at how the rank of the predicted tokens proceed for each generated token of the answer. More information about the dataset statistics can be found in Appendix C.

**Results:** Figures 4c and 4d show the first layer at which the correct fact recall token crosses a given rank threshold. The first observation is that all computations for fact recall tokens progress much slower ( layer 15) than function word categories from Case Study I ( layer 5). This means that recalling facts is not as easy as generating DET or ADP categories and requires much larger depth.

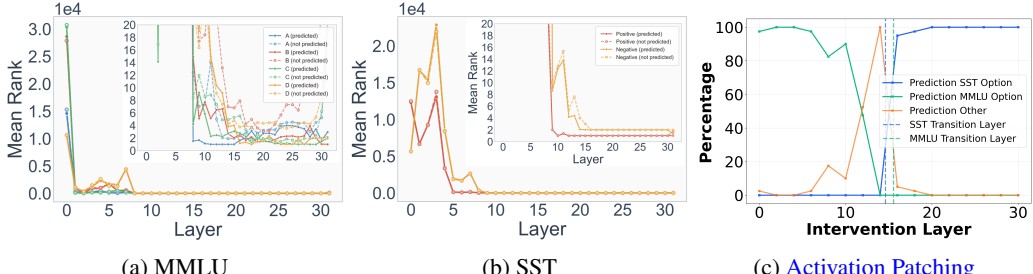

(a) MMLU        (b) SST        (c) Activation Patching

Figure 6: **Mean rank of answer tokens during constrained-choice downstream tasks.** As the model performs tasks like MMLU (response options: ABCD) and sentiment analysis ( response options: positive/negative), we track the mean rank of the options through the layers. Early layers of the model promote valid options choices as top-ranked candidates, whereas later layers reason between option choices (shown as zoomed-in figures in the top-right corner of each plot). This figure shows results for Pythia-6.9B. Results for other models and tasks follow similar patterns and can be found in Appendix C.1. (c) Activation patching experiments, as described in main paper, causally confirm the two-phases process.

We also see that the first appearance of the single-token answer happens much earlier into a model (layer 15 for Pythia-6.9B, layer 20 for Llama3-8B) compared to the first token of the multi-token facts (layer 25 for both models). This shows that recalling multi-token facts, especially the first token of a multi-token fact, is harder for the model compared to a single token fact. Predicting a single token fact only requires accurate construction of a single token, while predicting a multi-token fact also requires a model to anticipate the future tokens that complete fact recall accurately, which may make the recalling multi-token facts harder. Post-stability analysis where we measure the average layer at which predicted token crosses a given rank threshold for the *final* time are presented in Figure 17 in Appendix. We again see similar qualitative results at larger depth.

Critically, while computing a multi-token fact, *the earliest appearance of the first token in the multi-token fact requires the largest number of layers, but the subsequent tokens become top-ranked much sooner.* For instance, for a 3-token fact, the first token on average emerges around layer 27 for Pythia-6.9B, whereas the subsequent second and third tokens emerge around layers 20 and 12 respectively. This pattern represents another example of a graded use of model depth, where harder to do tasks, like predicting the first token in a multi-token fact, requires additional depth, while subsequent tokens that are conditionally easier to predict given the first token, appear at shallower depths.

**Early-Decoding Experiments:** To validate the information processing trajectories during multi-token fact recall, we perform early decoding and calulate Top-1 match rate at intermediate layers, similar to section 4.1. The results can be seen in Figures 5b and 5c, where the Top-1 match rate for subsequent tokens is much higher in early layers. This effect is especially stark when observing the second token in a 2-token fact and the third token in a 3-token fact for Pythia 6.9B.

## 4.3 CASE STUDY III: DEPTH USE DURING DOWNSTREAM TASKS

**Dataset and Methodology:** In this case study, we evaluate the internal prediction dynamics when the model performs four fixed-option downstream tasks - MMLU (Hendrycks et al., 2020), Sentiment Analysis (SST) (Socher et al., 2013), Natural Language Inference (NLI) (Dagan et al., 2005) and Paraphrase Dectection (MRPC) (Dolan & Brockett, 2005). We use a 4-shot set up, where the model is given four examples and is then asked to generate the answer. We then track the mean rank of the generated answer through the different layers of the model.

**Results:** The results can be seen in Figure 6. In these plots, we track the mean rank of the option choices through the different layers. We see a stark, two-step prediction pattern. In the early layers, the model identifies the valid option choices and gathers them within the top ranks of intermediate layer logits (option-collection). As a result, the early layers see massive decrease in the ranks of valid option choices for MMLU and SST datasets respectively. This step usually happens within the first half of the model. After the valid options are collected as top-ranked choices, the model reasons through the options in the remainder of the layers (option-reasoning). This can be seen within the

zoomed-in figures on the top-right corners of Figures 6a and 6b, where we zoom-in on the y-axis. For the MMLU task, the model constantly switches its top prediction, with some mid-layer bias for option A and late-layer bias for options C and D. For the sentiment task, the model defaults to "positive" as the label of choice all the way until the last layer. Similar option biases can be seen in other models (Appendix C.1), with extreme biases in smaller models (GPT2-XL).

Overall, we see that the model performs fixed-option downstream tasks in two phases - in the first phase the model collects the option choices as top-ranked tokens, and in the second phase it reasons between the option choices to generate an answer. In accordance with the complexity-driven usage of depth, the model performs the easier of the two tasks of collecting the valid option in the early layers, while it leaves reasoning to deeper layers.

### 4.3.1 CAUSAL INTERVENTION BY ACTIVATION PATCHING

**Setup:** To validate our assertion that fixed-option downstream tasks happen in two distinct phases as described above, we perform activation patching experiments (Meng et al., 2022). We consider two tasks, MMLU and SST, where SST is the source task and MMLU is the target task. For each layer, we take the activation vector corresponding to the generated token at a particular layer while processing an MMLU example, and replace it with the activation vector for the generated token during processing of an SST example from the same layer. The expectation is that if there are two distinct phases of option-collection and option-reasoning, then transfer of activation vectors during the option-collection phase should still give the model time to collect valid options, whereas transfer of activation vector during option-reasoning phase should generate one of the two options belonging to the SST categories.

**Results:** This is indeed what we see in Figure 6c. The y-axis of the figure measures percentage of times the predicted answer belongs to either one of the MMLU options (green line), or one of the SST options (blue line), or neither (yellow line). This plot shows replacement of activations for Pythia 6.9B. We see that if we replace the activation from SST to MMLU processing in early layers, which is the option-collection phase, the model is still able to collect the MMLU options to top ranks and produce one of the MMLU options as the final answer. However, if this transfer happens beyond the transition layers, which is the option-reasoning phase, the output belongs to one of the SST options. **This experiment clearly demonstrates the presence of a two-phase answering algorithm followed by the model during fixed downstream tasks.** Note that this activation replacement experiment is independent of the TunedLens probe. Results for other models show similar outcomes as shown in Figure 19.

We also define transition layers between the two phases using the TunedLens probe, which is the layer at which the model transitions from option-collection to option-reasoning mode. We define transition at the first layer at which all the top ranked tokens choices belong to the available options for the task. This means that for MMLU, the top four ranked tokens should belong to each of the four option choices in MMLU (A/B/C/D). Similarly, for SST, the top two ranked tokens should belong two each of the two options of the task (positive/negative). We see that the average transition layer when calculated using TunedLens matches very nicely with the transition happening in Figure 6c.

## 5 VERIFYING THE VALIDITY OF TUNEDLENS PREDICTIONS

The results presented in this paper rely on the predictions generated by TunedLens (Belrose et al., 2023). Thus, it is important to demonstrate that our conclusions reflect the internal representations of LLMs themselves rather than the biases introduced by TunedLens. We consider the possibility that the bias toward high frequency tokens was introduced by TunedLens itself and test it in two ways. First, we compare the probability of high-frequency tokens in each layer of TunedLens as compared to the final LLM layer and find them to be comparable. Second, we train our custom TunedLens by varying update frequencies of high-frequency tokens. We find that, even after reducing the update frequency of the most frequent token of a corpus by a factor of 1000, it still appears with an over-whelmingly high frequency in early layer top-ranked predictions. These tests show that early layer predictions in LLMs being dominated by high-frequency tokens is not a consequence of probe bias but rather a reflection of the information content in the early layer representations. Experimental details and additional experiments on the analysis of TunedLens faitfulness, as well as a comparison with LogitLens, can be found in Appendix D.

## 6 RELATED WORK

**Layerwise Decoding and Iterative Refinement:** The LogitLens framework (Nostalgebraist, 2020) allows us to interpret intermediate representations in the token space. Building on this, TunedLens (Belrose et al., 2023) trains lightweight affine probes to make intermediate predictions more faithful, especially for earlier layers, and explicitly frames transformers as performing iterative inference across depth. DecoderLens (Langedijk et al., 2023) extends the lens idea to encoder–decoder models and reports that specific subtasks are handled at lower or mid layers. Our work uses the TunedLens probe to explain the prediction dynamics happening during inferece using a "guess-then-refine" framework and an intelligent use of model depth. The idea of guess refinement was mentioned briefly in the original LogitLens blogpost (Nostalgebraist, 2020); however, that work provided no quantitative evidence of the refinement process, which is what we strive to do here.

Subsequent works to LogitLens also formalize saturation events, showing that top-1 token get locked-in before the last layer (Geva et al., 2022), and later layers continue to adjust margins and competitors, which was later generalized into a sequential lock-in process for top-k tokens (Liouba-shevski et al., 2024). Not all forward passes lead to saturation events and our work on the other hand focuses on pre-saturation events and explaining internal prediction dynamics during that regime.

**Factual Recall and Lookahead Computations:** Prior work shows that factual knowledge is stored within the MLP layers of LLMs (Geva et al., 2020; 2023; Meng et al., 2022) and also present evidence of lookahead planning in LLMs (Pal et al., 2023; Hernandez et al., 2023; Wu et al., 2024; Jenner et al., 2024; Men et al., 2024a). In our work, we tie these observations with depth-usage while performing multi-token fact recall by showing that the second and third token predictions of a multi-token fact requires less computation. Merullo et al. (2024) also study internal dynamics during fact recall. However, the specific mechanism studied in the paper shows that the early layers generate function arguments to retrieve factual information, which is very different from our work.

**Internal Mechanisms during In-Context Learning:** Many prior works have studied the internal mechanisms of in-context learning in LLMs (Xie et al., 2021; Dai et al., 2022; Von Oswald et al., 2023), but do not focus on how the ranks of labels get modified through the forward pass. Lepori et al. (2025) also explore the idea of refinement during the forward pass as more context accumulates, but focus on contextualization errors related to lexical semantics. Closest to our work is Wang et al. (2023) which shows that shallow layers store task semantics in label tokens, which is retrieved by later layers. Our work presents an orthogonal explanation of this process from the point of view of prediction dynamics and shows that shallow layers elevate the label tokens as top-ranked tokens, while deeper layers reason between them to perform the final selection.

**Dynamic-Depth Models and Early-Exiting:** Many recent (Gromov et al., 2024; Men et al., 2024b; Fan et al., 2024) and prior works (Bolukbasi et al., 2017; Huang et al., 2017; Elbayad et al., 2019) have shown that the entire depth of the model is not required for generation and have used this phenomenon to try and improve inference efficiency. Our work provides new insights for future early exiting work, including analysis showing that function tokens and subsequent fact tokens in multi-token fact recall are more likely to get decoded early.

## 7 CONCLUSION AND FUTURE WORK

In this paper, we explain the internal prediction dynamics of LLMs during inference by proposing a "*Guess-then-Refine*" framework. We show that early LLM layers promote high-frequency tokens to top-ranked predictions in early layers. These high-frequency tokens act as statistical guesses made by LLMs due to lack of enough contextual information about the input in the early layers. As further processing happens, these early layer guesses undergo massive refinement to eventually yield contextually appropriate tokens. We also show that LLMs leverage their depth flexibly in a task-dependent manner, by using early layers to perform easier tasks like predicting function tokens or identifying valid response options in a constrained-choice task setup, while using later layers for more complex processing like predicting content words, recalling facts, and reasoning. Together, our findings show that **LLMs are early statistical guessers and late contextual integrators that use their depth flexibly based on the complexity of predicted token or subtask.** Some avenues of future work includes a similar analysis for reasoning and chain-of-thought tasks and more recent reasoning models.

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

## A PREFIX CREATION

**How BASELINE is constructed.** We sample English Wikipedia (20220301.en). From each article we keep paragraphs that are single-line (no internal newlines) and reasonably long. For each kept paragraph we pick a random split point on a word boundary; the left side becomes the input prefix and the right side is discarded (next-token task uses only the prefix). We keep a prefix if it has at least 15 characters.

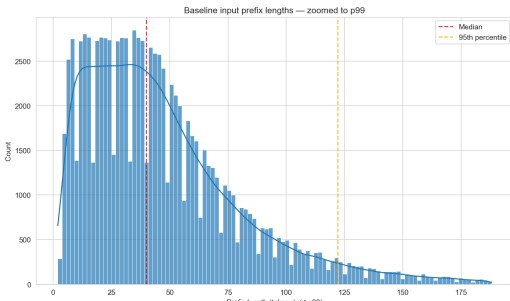

Figure 7: Distribution of BASELINE input prefix lengths (GPT-2 BPE tokens).

**What prefixes look like.** Below are raw examples (shortest / around median / longest / random).

**Shortest (by tokens)**
1 Volume production
2 Several different
3 Despite appearing

**Around median**
1 The Monmouth Hawks baseball team is a varsity intercollegiate athletic team of Monmouth University in West Long Branch, New Jersey, United States. The team is a member of the Metro Atlantic
2 The A.P. housing board and state government has provided a great housing colony with basic facilities to the poor people who lived in Hyderabad from many years and don't have own house so far.
3 The protein encoded by this gene is highly expressed in peripheral blood of patients with atopic dermatitis (AD), compared to normal individuals. It may play a role in regulating the resistance to apoptosis that

**Longest (by tokens)**
1 Ajike is recording in the studio when Ugo walks in with a date. ...
2 His parents, Franciszek Trabalski and Maria Trabalski, born Mackowiack, ...
3 Male, female. Forewing length 4.5 mm. Head: frons shining pale ochreous ...

**Random sample**

1    Despite the bad results in the Euro NCAP crash tests, statistics from the real

2    Glen Lake is a lake that is located north of Glens Falls, New York. Fish species present in the lake are rainbow trout, pickerel, smallmouth bass, largemouth bass, walleye, yellow perch,

3    The Malcolm Baronetcy, of Balbedie and

## B  FREQUENCY-CONDITIONED ONSET ABLATIONS

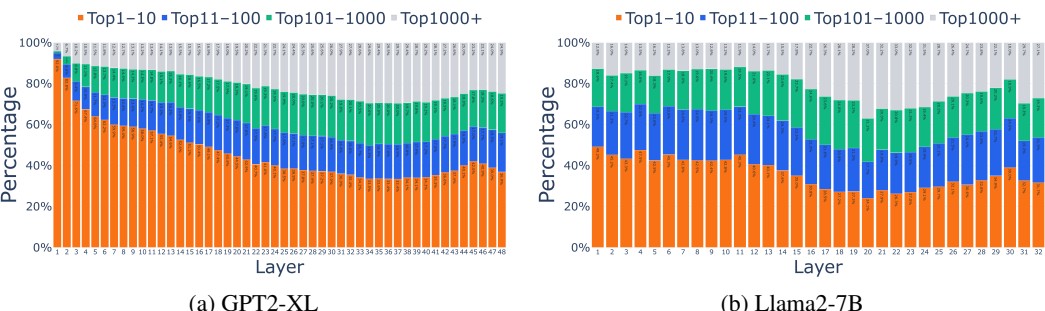

(a) GPT2-XL

(b) Llama2-7B

Figure 8: Evidence for Frequency-Conditioned Onset for GPT2-XL and Llama2-7B.

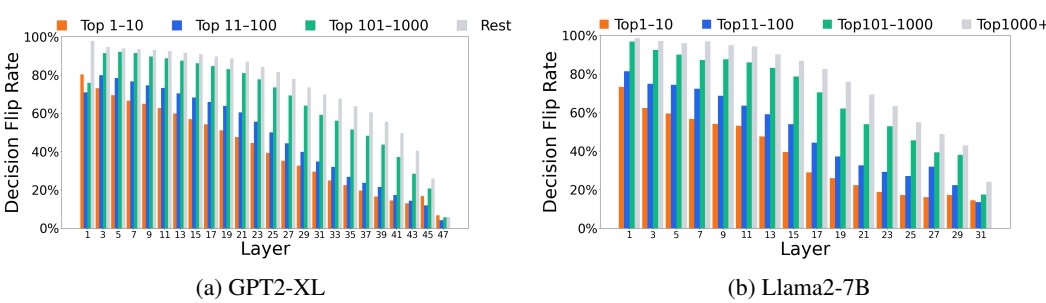

(a) GPT2-XL

(b) Llama2-7B

Figure 9: Early-layer predictions are provisional.

## C  COMPLEXITY-AWARE DEPTH USE : ADDITIONAL EXPERIMENTS

**Experiment Details for POS experiments:**  For each model and each POS category, the model is given 100k prefixes and the model predicts the next word given the prefix. The data is accumulated across all the prefixes.

**Experimental Details for Multi-Token Fact Prediction:**  The MQUAKE dataset (Zhong et al., 2023) originally contains 9,218 facts. We divide the MQUAKE dataset into examples where the answer tokens contain 1, 2 and 3 tokens. Additionally, we only use the prompts where the model generated the correct answer. The number of prompts used for each model and each type of fact is shown in Table 1.

| Model | 1t (pos1) | 2t (pos1) | 2t (pos2) | 3t (pos1) | 3t (pos2) | 3t (pos3) |
|---|---|---|---|---|---|---|
| GPT-2 XL | 1385 | 285 | 2190 | 98 | 480 | 631 |
| Pythia-6.9B | 1397 | 353 | 2085 | 409 | 846 | 924 |
| Llama 2-7B | 1705 | 820 | 2260 | 320 | 710 | 690 |
| Llama 3-8B | 1868 | 1086 | 2378 | 353 | 756 | 724 |

Table 1: Counts of distinct prompts by answer length and token position (1t/2t/3t = one/two/three-token answers; pos$k$ = $k$-th token in the answer).

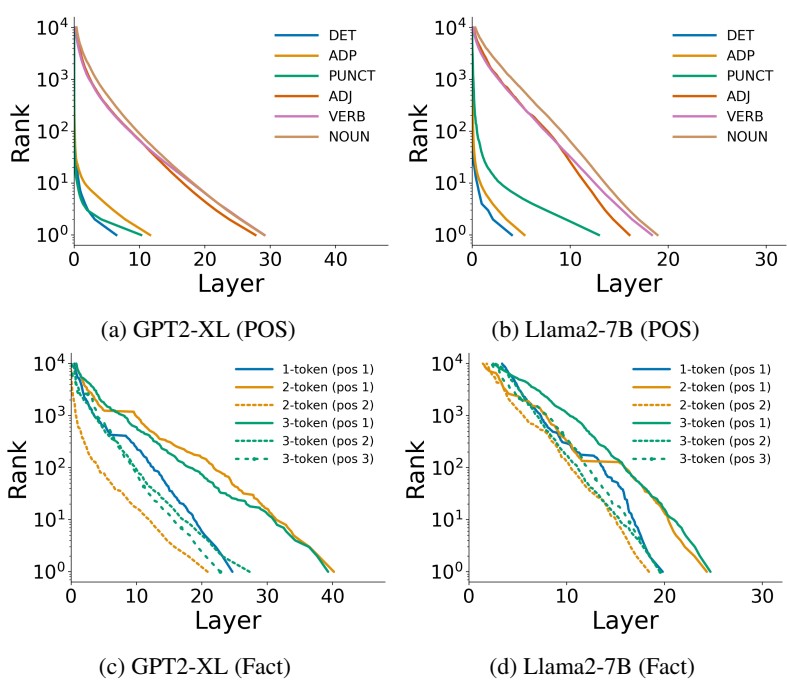

(a) GPT2-XL (POS)  (b) Llama2-7B (POS)

(c) GPT2-XL (Fact)  (d) Llama2-7B (Fact)

Figure 10: **Earliest crossing thresholds of predicted tokens by POS (top) and for multi-token facts (bottom).** We determine which token the model predicts at the final layer and then track the earliest layer (x axis) at which the TunedLens rank of that token crosses a given threshold (y axis). **(a, b)** Case Study I: When predicted tokens are grouped by POS category, tokens corresponding to function words (DET, ADP) and punctuation appear much earlier than content words (ADJ, VERB, NOUN). **(c, d)** Case Study II: We plot tokenwise results for fact recall responses that consist of 1 token, 2 tokens, or 3 tokens. E.g., 3-token facts consist of 3 tokens at positions (pos) 1, 2, and 3. In both models, the first tokens of multi-token facts appear much later than subsequent tokens.

## C.1 DOWNSTREAM TASK ABLATIONS

(a) GPT2-XL — MMLU

(b) GPT2-XL — MRPC

(c) GPT2-XL — NLI

(d) GPT2-XL — SST

(e) Llama-2-7B — MMLU

(f) Llama-2-7B — MRPC

(g) Llama-2-7B — NLI

(h) Llama-2-7B — SST

(i) Llama-3-8B — MMLU

(j) Llama-3-8B — MRPC

(k) Llama-3-8B — NLI

(l) Llama-3-8B — SST

(m) Pythia-6.9B — MMLU

(n) Pythia-6.9B — MRPC

(o) Pythia-6.9B — NLI

(p) Pythia-6.9B — SST

Figure 11: Mean rank by choice across datasets for multiple models

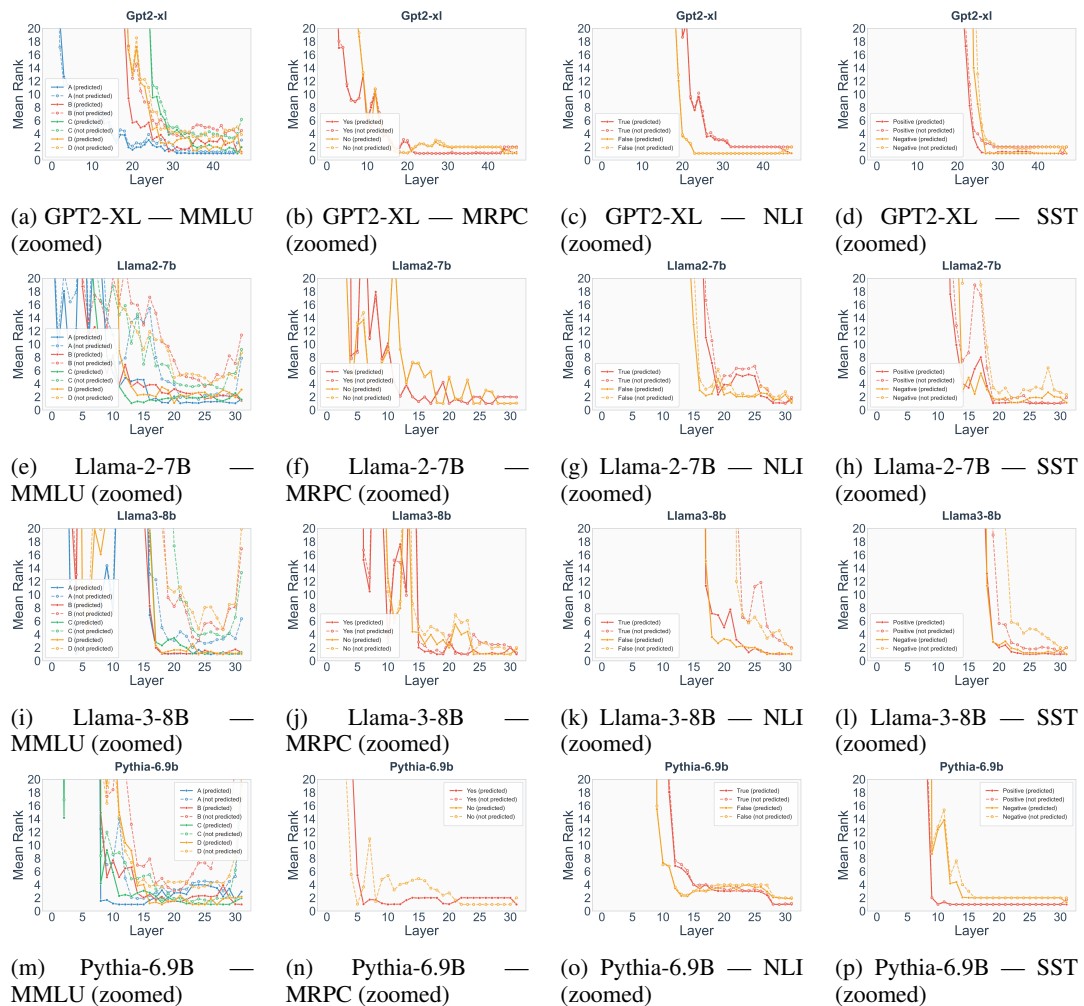

(a) GPT2-XL — MMLU (zoomed)

(b) GPT2-XL — MRPC (zoomed)

(c) GPT2-XL — NLI (zoomed)

(d) GPT2-XL — SST (zoomed)

(e) Llama-2-7B — MMLU (zoomed)

(f) Llama-2-7B — MRPC (zoomed)

(g) Llama-2-7B — NLI (zoomed)

(h) Llama-2-7B — SST (zoomed)

(i) Llama-3-8B — MMLU (zoomed)

(j) Llama-3-8B — MRPC (zoomed)

(k) Llama-3-8B — NLI (zoomed)

(l) Llama-3-8B — SST (zoomed)

(m) Pythia-6.9B — MMLU (zoomed)

(n) Pythia-6.9B — MRPC (zoomed)

(o) Pythia-6.9B — NLI (zoomed)

(p) Pythia-6.9B — SST (zoomed)

Figure 12: Mean rank by choice across datasets for multiple models (zoomed-in views

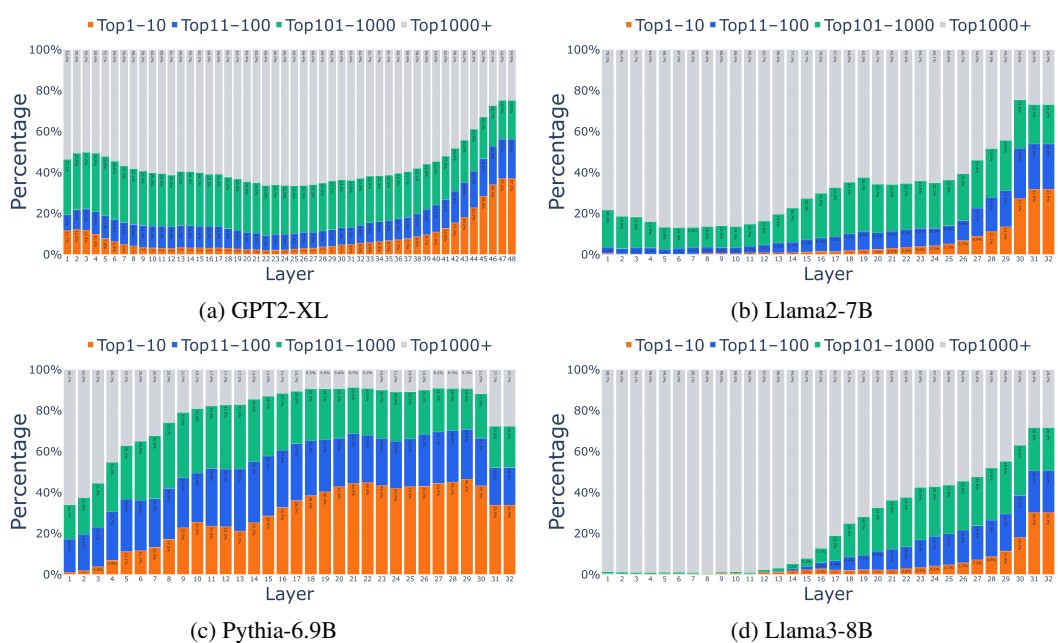

Figure 13: Top-ranked predictions in intermediate layers using LogitLens as the probe.

# D   TUNEDLENS VALIDITY

While we see that high-frequency tokens dominate the top-ranked predictions when using TunedLens, this is not the case when we use LogitLens to analyse the intermediate representations of the model. The top-ranked predictions for LogitLens can be seen in Figure 13. We see that high-frequency tokens are infact a minority in early layer top predictions, and infact the `Top1000+` bucket tokens dominate the early layer predictions using LogitLens. The likely reason for this is the inability of LogitLens to successfuly decode the early layer representations, which is made up of the final layernorm and unembedding matrix. This fact has been recorded in literature many times (Geva et al., 2023; 2022; Belrose et al., 2023), including the original LogitlLens blogpost (Nostalgebraist, 2020). Thus, it becomes hard to trust the top-ranked predictions of early layers using LogitLens. The most likely reason for the dominance of tokens from the `Top1000+` bucket in early layers is simply the fact that this bucket contains the largest number of tokens (of the order of tens of thousands).

While unfaithfulness of LogitLens to successfully decode early layer representations is well established, in this section we evaluate the faithfulness of TunedLens in decoding early layers. We specifically evaluate two things. Firstly, we check if TunedLens probe artificially assigns high probabiliy mass to high-frequency tokens. If this is the case, then this explains the occurence of high-frequence tokens as top-ranked proposals in early layers. We will see in section D.1 that this is not the case. Secondly, we check if the early layer predictions are a result of bias introduced during training of linear transformations. To do this, we train our custom TunedLens tranlsators by artificially changing the update frequency of high-frequency tokens. We find that this change does not impact the results in section 3, and the TunedLens results are a reflection of information content in early layers.

## D.1   PROBABILITY MASS ASSIGNMENT TO VOCABULARY TOKENS USING TUNEDLENS

To understand if high-frequency tokens being proposed in early layers using TunedLens, as shown in section 3, is an artificant of TunedLens probe or a reflection of the information content in early layers, we look at the average probability with which predictions are made in early versus late layers. To calculate the average probability of predictions in early layers, we use both the TunedLens and the LogitLens probe. We compare the average probability with which high-frequency tokens are

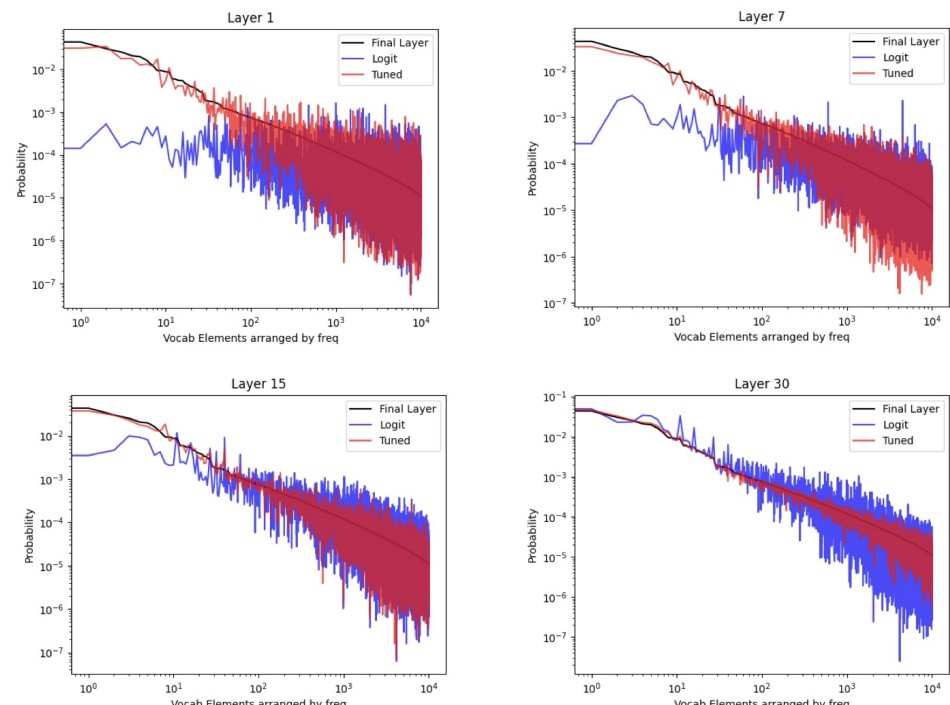

Figure 14: This plots the average probability of each vocabulary token at logits created at intermediate layers using LogitLens and TunedLens, when compared to their final layer probabilities. The x-axis contains the list of vocabulary tokens arranged in decreasing order by frequency. This mean first element on the x-axis corresponds to the most frequent token in the vocabulary, the second element is the second most frequent and so on. The y-axis represents the average probability with which that token gets predicted at a certain layer using TunedLens (in red) and LogitLens (in blue), when compared to the average prediction probability of the token at the final layer (in black). The plots are shown for Pythia-6.9B model for 4 intermediate layers.

predicted in early layers versus the lower frequency tokens. The aim of this experiment is to see if TunedLens assigns an unusually larger probability mass to high-frequency tokens.

**Methodology:** We look at the next-token prediction task, as in section 3. For an input prefix, the model makes a prediction. We then track the probability vectors corresponding to the hidden representations at each layer. These probability vectors are created using both TunedLens and LogitLens.

For a given vocabulary $V$, we first arrange the vocabulary elements in a decreasing order by their frequency and call this list $V_s = \{v_1, v_2, \ldots, v_{|V_s|}\}$. Thus, the first vocabulary element of this list, $v_1$, is the most frequent token of the vorpus, and so on. Then, for each element $v_i \in V_s$, we find the probability with which that token gets predicted in each of the intermediate and final layer, and find the average of this probability across the list of input prefixes that are sent to the model.

For example, let's say we send in $N$ prefixes $x_1, x_2, \ldots x_N$ into a model for this experiment. Then, for each prefix, we get a probability vector at each layer $l$ corresponding to either TunedLens ($p_l^{tuned}$) or LogitLens ($p_l^{logit}$). Then, for each layer, we calcualte the average probability of a token $v_i$ as $1/N \sum_{j=1}^{j=N} p_l^{lens}(v_i)$ . We compare the average probability for each token at each layer using LogitLens and TunedLens with the average probability of each token at the final layer.

**Results:** When using TunedLens to analyse early layer representations, we see a lot of high-frequency tokens being predicted at intermediate layers as top-ranked tokens (section 3). With this experiment, we check if TunedLens is assigning unnaturally high probability mass to high-frequency tokens. The results can be seen in Figure 14. The x-axis contains the list of vocabulary tokens arranged in decreasing order by frequency. This mean first element on the x-axis corresponds

to the most frequent token in the vocabulary, the second element is the second most frequent and so on. The y-axis represents the average probability with which that token gets predicted at a certain layer using TunedLens (in red) and LogitLens (in blue), when compared to the average prediction probability of the token at the final layer (in black).

*We see that the average prediction probability of TunedLens matches the average prediction proability of the final layer distribution, while LogitLens highly underestimates the probability mass for high-frequency tokens in early layers*. This brings out the lack of faithfulenss of LogitLens in succesfully decoding the early layer representations. As we go deeper into the model, the Logitlens decoding becomes closer and closer to the final layer distribution and slowly becomes more faithful. Thus, our experiments show that TunedLens does not place additional probability mass over high-frequency tokens.

### D.2 Training Custom TunedLens

In this paper, we analyze the intermediate representation of LLMs in the token space using TunedLens (Belrose et al., 2023). We use TunedLens as our probe of choice since it is known to produce more faithful intermediate predictions than Logit Lens, particularly in early layers (Belrose et al., 2023). TunedLens is trained to minimize the KL-divergence between the final layer probability distribution and an intermediate layer probability distribution. When training a probe for layer $l$, let the input token representation to the model be $h$, and intermediate hidden representation at layer $l$ be $h^l$. Then, the TunedLens objective can be written using the following equation:

$$\underset{A_\ell, b_\ell}{\operatorname{argmin}} \mathbb{E}\left[D_{KL}\left(f_\theta(h^l) \,\|\, \text{TunedLens}(h^l)\right)\right] \tag{4}$$

where $f_\theta(h)$ represents the probability distribution for input representation $h$ at the output of the model, $A_l$ and $b_l$ represent the parameters of the TunedLens probe following equation 2, and TunedLens($h^l$) represents the probability distribution at an intermediate layer $l$ using the TunedLens probe.

KL-divergence is known to exhibit frequency bias, that is, it has the tendency to be dominated by high-frequency events. This means that higher frequency tokens may have a greater say in minimizing the loss in equation 4 compared to lower frequency tokens. In section 3.1, we show that most early layer predictions using TunedLens correspond to high-frequency tokens, which could have two potential explanations. Firslty, that the early layer representations actually contain information content corresponding to high-frequency tokens, in which case the results in section 3.1 present an accurate functioning of LLMs. The alternative is that the results in section 3.1 are an artifact of the KL-divergence frequency bias and the TunedLens probe is only able to learn effective transformations for high-frequency tokens.

We first answer this question qualitatively by pointing out that if predicting high-frequency tokens were an artifact of the TunedLens probe, this would be true for all layers since the TunedLens transformations are trained for all layers independently using the same loss function. Instead, we observe high-frequency tokens being predicted only in early layers. This indicates that the prominence of high-frequency tokens in early layers is because of the information content of the early layer representations. To make sure this intuition is correct, we carefully train our own TunedLens probes by reducing the frequency of some high-frequency tokens during training. We do this by masking the KL-loss of high-frequency tokens.

**Methodology:** We train two versions of TunedLens probes using the original codebase (Belrose et al., 2023). We first create a baseline version where we retrain the TunedLens probe from scratch for each model to reproduce the results from Belrose et al. (2023). We then train a modified version of TunedLens where we select a high-frequency token belonging to the `Top1-10` bucket, and change its update frequency in the KL-loss to match the update frequency of a token from the `Top101-1000` bucket. Specifically, we do this for the token "*the*" since its commons for all tokenizers, whose corpus frequency is around 4-5% for all models. We then mask the contribution of this token in the KL-loss in equation 4 and bring it down by a factor of 1000, which effectively means that the probe is updated for the "*the*" token at a frequency similar to a token belonging to the `Top101-1000` bucket. We then analyze the change in frequency with which this token appears as a top-1 prediction using our modified TunedLens.

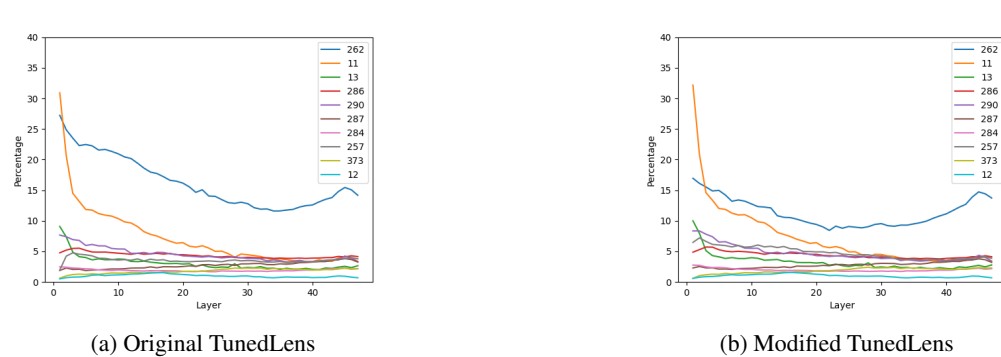

(a) Original TunedLens                                    (b) Modified TunedLens

Figure 15: We train our custom TunedLens by masking the update frequency of the token 262 ('the'), the second most frequent token in early layers but the most frequent token during final prediction. We reduce the update frequency of token 262 by a factor of 1000. Figure 15b shows the frequency of the top-10 tokens for our modified TunedLens, in comparison to the original model in Figure 15a. The example in the figure is for Pythia-6.9B.

**Results:** The results for the probing experiments are shown in Figure 15. During training, we bring down the update frequency of the target token by a thousandth of its original value, making it match the `Top101-1000` bucket. Yet, we see that the token "the" occurs as the top-1 TunedLens prediction in the early layers with a very large majority, and is still the second most frequently predicted tokens in early layers. Meanwhile, other tokens in the `Top101-1000` bucket are predicted with much lower frequency in early layers. This shows that the early-layer predictions of TunedLens probe corresponding to high-frequency tokens actually represents the information content in the intermediate representations of early layers rather than probe bias.

# E  POST-STABILITY PLOTS DEPTH USAGE EXPERIMENTS

This is an ablation for Case Study I in section 4.1. Here, we determine which token the model predicts at the final layer and then track the layer (x axis) at which the TunedLens rank of that token crosses a given threshold (y axis) for the final time. Differentiating from section 4.1, we plot the last-time a rank threshold is crossed by the token. Thus, if the predicted tokens lies point (x,y) on the plot, this means that on average after layer x, the predicted token stays within the top-y tokens and never crosses that top-y rank. This shows the average post-stability statistics. Figure 17 contains the same experiment for fact prediction using the post-stability last crossing metric counterpart for section 4.2.

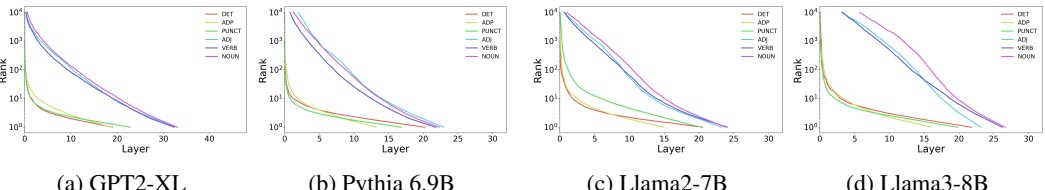

| (a) GPT2-XL | (b) Pythia 6.9B | (c) Llama2-7B | (d) Llama3-8B |

Figure 16: **Post-stability last-crossing of rank thresholds of predicted tokens by POS category** We determine which token the model predicts at the final layer and then track the layer (x axis) at which the TunedLens rank of that token crosses a given threshold (y axis) for the final time. This is an ablation for Case Study I in section 4.1.

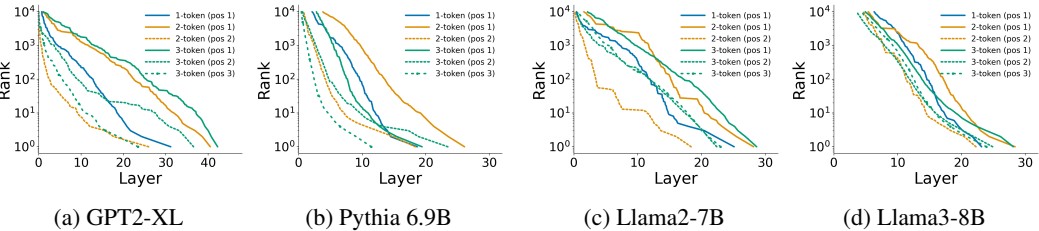

| (a) GPT2-XL | (b) Pythia 6.9B | (c) Llama2-7B | (d) Llama3-8B |

Figure 17: **Post-stability last-crossing of rank thresholds of predicted tokens by Fact prediction** We determine which token the model predicts at the final layer and then track the layer (x axis) at which the TunedLens rank of that token crosses a given threshold (y axis) for the final time. This is an ablation for Case Study I in section 4.2.

# F   EARLY-DECODING ABLATIONS FOR DIFFERENT MODELS

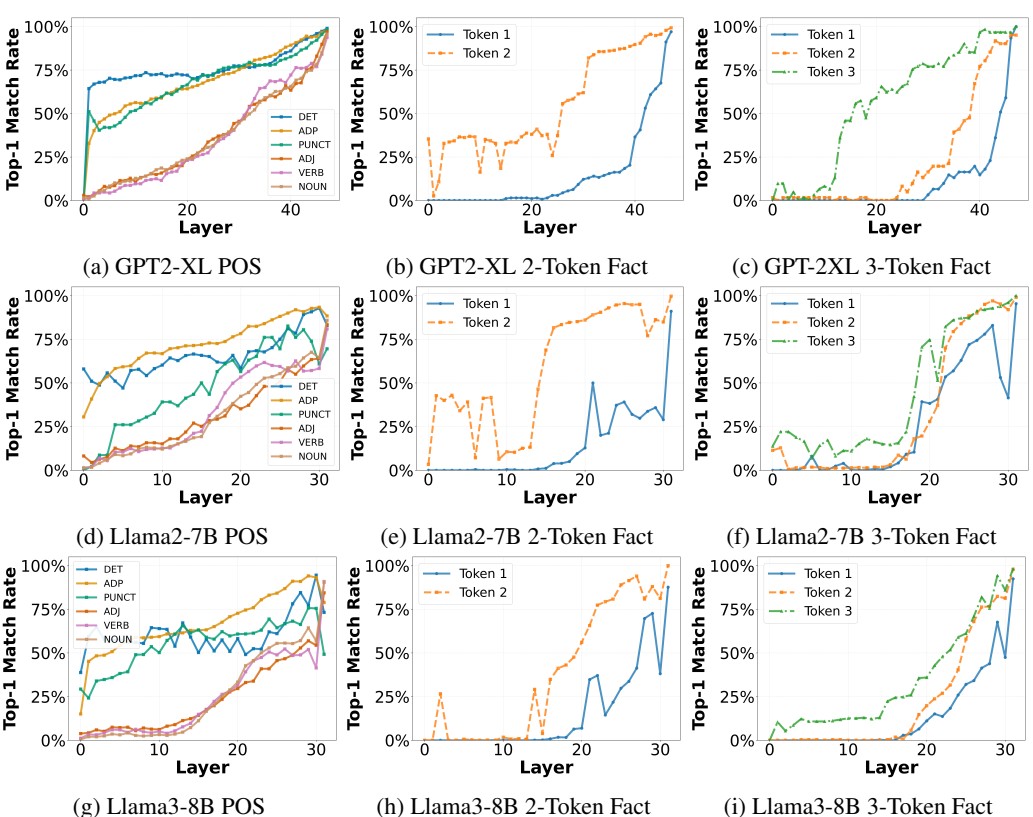

(a) GPT2-XL POS          (b) GPT2-XL 2-Token Fact          (c) GPT-2XL 3-Token Fact

(d) Llama2-7B POS        (e) Llama2-7B 2-Token Fact        (f) Llama2-7B 3-Token Fact

(g) Llama3-8B POS        (h) Llama3-8B 2-Token Fact        (i) Llama3-8B 3-Token Fact

Figure 18: **Early-decoding experiments.** To test our hypothesis that easy to predict token categories indeed finish processing earlier a larger proportion of times, we find the Top-1 match rate for an intermediate layer decoded token with the final prediction. The decoding is done via TunedLens. This contains ablations for other models in continuation to Figure 5.

# G ACTIVATION PATCHING FOR DOWNSTREAM TASKS

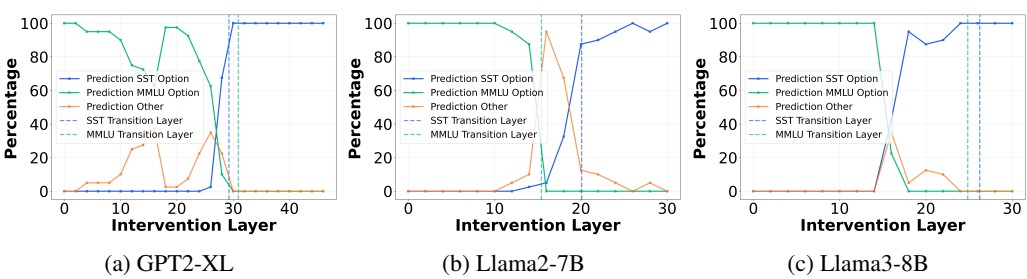

(a) GPT2-XL      (b) Llama2-7B      (c) Llama3-8B

Figure 19: **Activation Patching**

We define **transition layers** between the two phases using the TunedLens probe, which is the layer at which the model transitions from option-collection to option-reasoning mode. We define transition at the first layer at which all the top ranked tokens choices belong to the available options for the task. This means that for MMLU, the top four ranked tokens should belong to each of the four option choices in MMLU (A/B/C/D). Similarly, for SST, the top two ranked tokens should belong two each of the two options of the task (positive/negative).

The transition layer nicely predicts the transition between option-collection and option-reasoning phases for GPT2-XL, Pythia 6.9B and Llama2-7B. However, it does not prediction the transition in Llama3-8B as shown in Figure 19 (c). This can be explain when looking at the top ranked options at intermediate layers for Llama3-8B. We see that in Figure 20. We see that the top choices in middle layers for Llama3-8B have 'neutral' as one of the top choices around the transition layers, however 'neutral' is not one of the options. For MMLU, it is able to promote 3 out of the 4 options into top4 correctly as you said. And sometimes one of the options is lowercase. This process happens much earlier but exact options get promoted much later, which delays the TunedLens prediction of transition layer.

```
model.layers.0 0 [' (', ' and', ' ', ' to']
model.layers.1 1 [' we', ' We', ' you', ' ']
model.layers.2 2 [' The', ' ', ' "', 'The']
model.layers.3 3 [' Is', ' "', ' ', ' The']
model.layers.4 4 [' A', ' The', ' "', ' someone']
model.layers.5 5 [' The', ' ', ' "', ' A']
model.layers.6 6 [' The', ' "', ' A', ' D']
model.layers.7 7 [' \\', ' **', '\r\n', 'times']
model.layers.8 8 [' _', ' "', ' ', ' $\\']
model.layers.9 9 [' \n', ' \\\\', ' "', ' \n\n']
model.layers.10 10 [' (', ' "', '3', '6']
model.layers.11 11 [' \n', ' _', ' ', '6']
model.layers.12 12 ['110', '1', '38', '6']
model.layers.13 13 [' _', '3', ' ', ' [']
model.layers.14 14 [' _', '3', ' ', ' third']
model.layers.15 15 [' _', '3', ' ', '2']
model.layers.16 16 [' _', ' **', ' ', ' A']
model.layers.17 17 [' _', ' B', ' ', ' **']
model.layers.18 18 [' B', ' ', ' b', ' A']
model.layers.19 19 ['B', ' B', ' ', ' A']
model.layers.20 20 ['B', ' B', ' ', ' A']
model.layers.21 21 ['B', ' B', ' ', ' A']
model.layers.22 22 ['B', ' B', ' ', ' A']
model.layers.23 23 ['B', ' B', ' ', ' A']
model.layers.24 24 [' B', ' C', ' A', ' D']
```

```
model.layers.6 6 [' neighborhood', ' ']
model.layers.7 7 [' \n', ' ']
model.layers.8 8 [' \n', ' ']
model.layers.9 9 [' \n', ' ']
model.layers.10 10 [' \n', ' ']
model.layers.11 11 [' \n', ' ']
model.layers.12 12 [' \n', 'eger']
model.layers.13 13 [' \n', ' \n']
model.layers.14 14 [' \n', ' ']
model.layers.15 15 ['289', ' -']
model.layers.16 16 [' \n', ' ']
model.layers.17 17 [' \n', ' love']
model.layers.18 18 [' ', ' neutral']
model.layers.19 19 [' positive', ' neutral']
model.layers.20 20 [' positive', ' neutral']
model.layers.21 21 [' positive', ' neutral']
model.layers.22 22 [' positive', ' neutral']
model.layers.23 23 [' positive', ' neutral']
model.layers.24 24 [' positive', 'positive']
model.layers.25 25 [' positive', ' positive']
model.layers.26 26 [' positive', ' positive']
model.layers.27 27 [' positive', ' neutral']
model.layers.28 28 [' positive', ' neutral']
model.layers.29 29 [' positive', ' negative']
```

(a) Llama3-8B MMLU      (b) Llama3-8B SST

Figure 20: **Example**

## H    FACT RECALL CORRECT VS INCORRECT ANSWERS

In this section, we do a comparison in the internal processing of tokens during fact recall, in continuation to Case Study II (section 4.2). Here, the model is given a question as prefix and is asked to generate an answer. The reason we focus on the prediction dynamics when the model generated the correct answer is because it easier to recognize that the model generated a fact, and the span of the fact. This is explained by an example below.

If given input to the model is the prefix - *"The capital of USA is"*, the model can continue this with broadly three types of generations:

1. **Correct Fact Recall** - Here, the model generates the correct answer, which would be "Washington D.C.".

2. **Incorrect Fact Recall** - Here, the model generates an incorrect answer, but the answer corresponds to a fact recall. An example of an incorrect fact recall would be "New York".

3. **Incorrect Non-Fact Phrase Generation** - Here, the model can continue the input "The capital of USA is" by generating the phases *"the best city in the world"*.

Now, if we only look for the correct answer in model generation, we know exactly how many tokens the model needs to generate. This allows us to let the model generate that many tokens, and then compare with the correct answer.

If the answer is incorrect, we don't know where the answer stops. For example, let's say the ground truth has 5 tokens in it, as in the example where the correct answer is "Washington D.C.", where the 5 tokens are ['Washington', 'D', '.', 'C', '.']. Let's say the model generates the answer incorrectly, which is "New York", but if we let it generate 5 tokens as in ground truth, then the model will generate three extra tokens that do not correspond to fact recall. Also, when the model is incorrect, it could be generating a non-factual phrase like "the best city in the world", and there would be no simple way to differentiate between the two or find the fact boundary.

**Method:** We evaluate whether generation is a fact and the span of the fact by using LLM-as-a-Judge. We use Gemini 2.5 as the judge and follow the following process:

- **Step-1:** When the model is incorrect, we employ LLM as a judge to first spot if the incorrectly generated answer corresponds to a incorrect factual generation, or a general non-factual phrase generation. We only consider the case where the model does factual recall, since we want to study the model behavior in fact recall setting.

- **Step-2:** In cases of incorrect fact generation, we let the model generate 5-6 tokens. We then ask our Judge to identify the span of the fact in the generation. For example, when the input is "The capital of USA is", the model could do an incorrect fact recall and answer "New York and it is awesome", but the fact recall is happening only during the first 2 words. So our Judge identifies the span of fact recall in the incorrect factual generation.

Once we find the span of the incorrect factual generation and verify the generation corresponds to the fact recall process, we are able to track the rank of incorrect factual predictions. This can be seen in Figure 21. We see that the pattern of subsequent tokens in a multi-token fact requiring fewer layers holding for 2 token facts for all models, and the third token progressing faster than the first token holds for all models as well. We additionally find that multi-token facts no longer require larger number of layers on average when compared to single-token facts. We hypothesize that this happens since the model is recalling non-ground-truth facts, the recall mechanism takes less effort to produce the first token in a multi-token fact recall setting.

These experiments however do not contract our results. Our cleanest result holds when the model is able to do fact recall correctly, where multi-token facts require larger depth, and first of the multiple tokens requires larger depth of processing than subsequent tokens. The observation that this pattern is slightly different during incorrect factual recall actually opens up new avenues for further exploration of using this as a metric to study the difference in mechanisms between correct and incorrect factual recall.

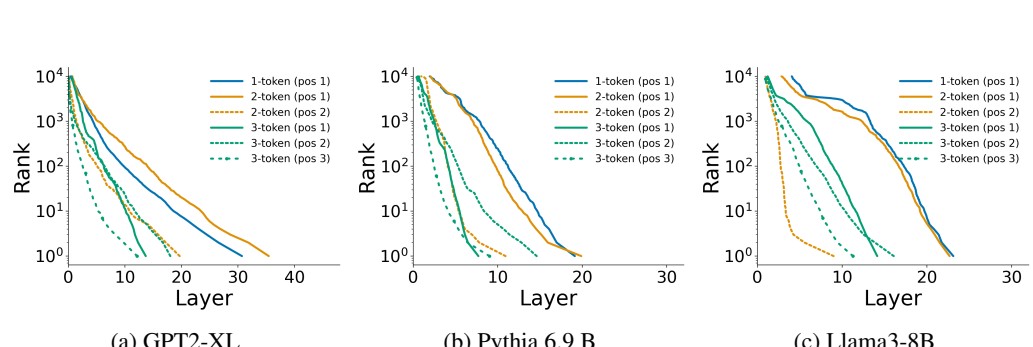

(a) GPT2-XL      (b) Pythia 6.9 B      (c) Llama3-8B

Figure 21: Incorrect Fact Recall

