# OpenReview forum: "How Do LLMs Use Their Depth?"
_ICLR.cc/2026/Conference — Submitted to ICLR 2026_

### Official Review · Reviewer_6FJo · 2025-10-16

**Soundness:** 3
**Presentation:** 4
**Contribution:** 2
**Rating:** 6
**Confidence:** 4

**Summary:**

This work provides a variety of empirical studies demonstrating that different tasks, and even different token generations within a task (based on part-of-speech), require different model-depths and that LLMs are able to dynamically assign depth. Results show that as data propagates through the LLM, the LLM's prediction goes from a training-frequency-motivated prediction to a more context-specific one.

**Strengths:**

The paper is written in a clear and concise way. Motivations, rationales, and design decisions are always outlined in advance. Analyses of results are always clearly given. Figures are easy to read. The findings suggest an intuitive and useful interpretation for some underlying mechanisms of the LLM, which may prove especially useful for early exiting work.

**Weaknesses:**

1. At times, "refinement" is used when really only "change" is demonstrated. For example, Figure 3 demonstrates that early predictions are often changed. This is described as demonstrative of refinement in the paper, but if early predictions are changed incorrectly or not correctly (i.e. are originally correct and flipped to incorrect, or are originally incorrect and changed to a different incorrect token), this is not refinement but simply an artifact of low-confidence in early-layer predictions. Figure 3 also shows a similar trend for all three token bins, making this the only real drawable conclusion. Showing specifically "correct" flips might be more informative.
2. Some valuable and apparent continuations are missing (e.g. layer-wise behavior analysis for correct vs. incorrect final predictions, identifying a priori internal indicators of finality of prediction/high confidence-ness).
3. Some experimental design is questionable: it is never explained why only correctly answered questions are analyzed in Case Study 2.
4. In general, the notion that the LLM intelligently assigns depth is invoked. This is somewhat problematic because the LLM continues to process these tokens through the full depth of the model. While most of the time a mid-layer high-confidence prediction may not be flipped, it on occasion will be and these flips can contribute to small changes overall performance metrics.

**Questions:**

1. The work claims and aims to demonstrate that the LLM has an innate sense of depth-requirement for predictions. If the LLM knows this a priori as is implied by the language in the paper, could predictor signals in the layer-wise hidden states of the model be found? This would be greatly beneficial from a cost-saving perspective, as we could early-exit the LLM, avoiding at times the majority of the forward-pass computation.
2. Why do you only show results for correctly-answered problems in Case Study 2
3. Does the rank-threshold behavior shown apply similarly to correctly and incorrectly answered questions? Could we potentially identify problematic LLM answers in advance by noting, for example, unusually late threshold-passing?
4. How frequently do predictions enter the top-k threshold and then leave it later? Are these cases more typically correct or incorrect predictions?
5. Could the part-of-speech distinction in layer-wise confidence progression be a simple matter of some parts of speech having much fewer options than others? For example, there are only a handful of punctuation marks, but thousands of verbs. Presumably, the LLM has a strong sense of grammar and understands the necessary part of speech at each token prediction well in advance of identifying a high-confidence prediction. Showing how often a token's predicted part of speech flips may be interesting.

---

> ### Author Response · Authors · 2025-11-27
>
> We thank the reviewer for taking the time to review our work and providing valuable feedback. We have updated our paper with new experiments, specifically with causal interventions that strengthen the claims of our work (which we summarize as a general comment to all reviewers). We have highlighted the major changes in the new content with blue text to reduce the cognitive load on the reviewers. We will refer to our latest uploaded version when referring to different sections, figures or lines of the paper.
> Below, we provide a response to the comments made by the reviewer.
>
> **WEAKNESS SECTION:**
>
> 1. We thank the reviewer for raising this point for clarification. In the entirety of section 3.2, we compare intermediate layer top-1 predictions against the final prediction, and count early layer predictions that get modified into the final layer. Therefore, we believe we are calculating exactly the metric the reviewer wanted to see. We have modified the language of the paper to make this more explicit (lines 243-248).
>
> 2. (combined response to weakness 2 and 3) -  We have added an analysis of incorrect predictions to complement our analysis of correct predictions (Appendix H). There, we also provide a justification for focusing on correct predictions in the main paper. The main reason for keeping only correct options was that it allowed us to cheaply figure out if the generated answer constitutes  fact recall and the start and end point of the recalled fact. If we don’t rely on ground-truth, we are unable to recognize both these things without using a labeller. To address the reviewer’s concern, we now also  analyze the scenarios where the model generates incorrect answers in Appendix H by using LLM-as-a-Judge (Gemini 2.5) to figure out instances of incorrect factual recall and answer spans. We find that the qualitative results remain the same with minor differences (we refer the reviewer to Appendix H for more details).
>
> 3. Answered above
>
> 4. We appreciate the reviewer pointing out this implication and we agree with the reviewer that our experiments do not indicate that LLMs have an a prior innate sense depth requirement. We have therefore modified such language in the paper, specifically replacing the phrase “complexity-aware” to “complexity driven” (For example, line 102, title of section 4 and subsequent sections). However, our experiments do indicate that the depth required by the model to perform a task is correlated with task complexity, as shown in section 4. The exact manifestations of these correlations are presented in our work. We have also added some additional experiments (Figure 5, Figure 6c) to strengthen these claims. The newer experiments in Figure 5 and 6c show causal interventions where our findings in section 4 manifest different experimental settings.
>
> **QUESTIONS:**
>
> 1. Answered in weakness #4
>
> 2. Answered in weakness #2. We refer the reader to appendix H for a detailed explanation of why we chose to do this. Short answer - we made this choice for computational efficiency.
>
> 3. Answered in weakness #2.
>
> 4. This is an interesting question, and we now perform a post-stability analysis for this, presented in Appendix E. The aim is to look at the rank progression of the predicted token, but instead of recording the first entrance into top-k tokens, we record the last entrance to the top-k tokens. Essentially, we record the entrance to top-k tokens after which the token never leaves top-k for a given value of k. Figures 16 and 17 show the results for that in Appendix E. Qualitatively, we see the same results as the first entrance metric, but the pattern is just shifted deeper into the model, which is expected.
>
> 5. We thank the reviewer for this insightful comment. Function-word categories like DET, ADP, and PUNCT are good examples of low-entropy (easier to predict) tokens, they are composed of very high-frequency items (high unigram probability) and they come from relatively small syntactic classes, as the reviewer correctly pointed out, both of which plausibly contribute to making them easier to predict. Fully disentangling the roles of frequency and class size is an interesting nontrivial problem, best suited for future work. We want to highlight that our goal in the POS analysis is to argue that certain tokens are easier for the model to predict, and that these tend to be resolved earlier in depth, which is also the larger theme of our work.
>
>
> We again thank the reviewer for their comments and interesting discussion. We hope that our responses answer the clarifications asked by the reviewer, and our additional experiments give the reviewer increased confidence in the validity of our results. We gently request the reviewer to take these results into consideration and update their scores as they find appropriate. We are also happy to engage in further discussion and answer any questions.

---

### Official Review · Reviewer_EivB · 2025-11-01

**Soundness:** 4
**Presentation:** 4
**Contribution:** 4
**Rating:** 8
**Confidence:** 4

**Summary:**

How LLMs use their depth for various tasks has been a key mystery. This paper does a systematic analysis across 3 categories of tasks to try and understand how LLM predictions vary as we go deeper into the stack. Observations highlight a “guess-and-refine” mechanism, wherein the model guesses high frequency tokens early on. In fact, another interesting (and intuitive) observation is that harder tasks need more depth. The paper is well structured, neatly written and the claims are well grounded.

**Strengths:**

- The three categories of tasks analysed not only show the breadth of the observations and the study, but rather also help clarify the observations from other categories.
- The claims made are very clear and well supported. No extravagant claims are present.
- The task difficulty vs layer prediction analysis is super nice.

**Weaknesses:**

- Any analysis on reasoning/CoT tasks would further significantly improve the quality and scope of manuscript.
- The manuscript would benefit from a more detailed discussion on the results, what implications they might have for practitioners or any suggestions or algorithms to improve the performance (say prediction depth) on hard tasks, based on the observations in the manuscript.

**Questions:**

See the weakness section

---

> ### Author Response · Authors · 2025-11-27
>
> We thank the reviewer for taking the time to review our work and providing valuable feedback. We have updated our paper with new experiments, specifically with causal interventions that strengthen the claims of our work (which we summarize as a general comment for all reviewers). We have highlighted the major changes in the new content with blue text to reduce the cognitive load on the reviewers. We will refer to our latest uploaded version when referring to different sections, figures or lines of the paper.
>
> Below, we provide a response to some of the comments made by the reviewer in the weakness section.
>
> 1. We agree that a detailed analysis of LLM depth-related representational dynamics in a reasoning context would be an important advance. However, due to the complexity of the question, it deserves its own paper (we have started some work in that direction). We now mention this future direction in the paper (section 7).
>
> 2. Thank you for this suggestion. The most immediate implication of our results is in the domain of early exiting, or more generally, using depth more efficiently. We have added several causal manipulations that illustrate this. First, we show differential effects of early exiting on different token types (Figure 5). We also discuss the implications of our work for early exiting  (lines 358-361, 523-525 and see the corresponding section in Related Work). Finally, we add an activation patching experiment, which validates our initial conclusion that the model settles on the format of the response first and only then on the content (section 4.3.1).
>
> We again thank the reviewer for their comments and feedback. We believe that our responses and additional experiments address comments raised by the reviewers. We gently request the reviewer to take these results into consideration and update their scores if they find appropriate. We are also happy to engage in further discussion and answer any other questions.

---

### Official Review · Reviewer_1K5T · 2025-11-01

**Soundness:** 1
**Presentation:** 2
**Contribution:** 2
**Rating:** 2
**Confidence:** 3

**Summary:**

The paper investigates how LLMs use depth at inference and advances a “guess-then-refine” account: early layers produce frequency-biased provisional guesses, while later layers integrate context to finalize tokens. Using TunedLens to decode intermediate states, it measures onset and flip-rate metrics across multiple models and tasks (POS, fact recall, multiple-choice), yielding an empirical picture of layerwise dynamics with implications for early exit and routing.

**Strengths:**

* The paper propose a simple, direct “guess-then-refine” view of depth with lightweight metrics. It provides analysis of TuneLens method results.
* Experiments cover multiple model families and tasks with transparent data.
* The findings might have a practical usage beyond interpretability (e.g. early-exit and routing in LLM systems).

**Weaknesses:**

* The probe by tunelens is trained to mimic the final distribution, so agreement with the final layer is not independent evidence and may imprint the reported pattern.
    - If the layer embeddings are matched toward that of the last, naturally, it would generate related tokens and patterns, but that is from the affine mapping, but not the model.
    - I think this is the most critical issue.
    - The authors can try add a probe that does not target the final distribution (or potentially combine with LogitLens) and compare with activation patching to test causality.
* No causal validation - decoded ≠ computed: All claims come from decoding intermediate states rather than causal tests. A probe can read out patterns that are present but not functionally responsible for the final prediction. So the curves may show “guess-then-refine” because it’s easy to decode, not because those layer states actually cause the model’s choice. I understand this is a general limitation in interpretability work, but the lack of causal evidence—and with possible probe-driven bias—makes the findings not convincing.
* Fact-recall analyses keep only correct cases, which induces survivorship bias and makes the depth story look cleaner than it is. Include failure-conditioned analyses and compare onset and flip rates when the model is wrong.
* The first-crossing onset metric is brittle under non-monotonic ranks and can mark spurious early onsets. Report stability-based onsets that require persistence in top-k and include threshold sensitivity analyses.
* Frequency buckets are built on English Wikipedia with model-specific tokenizers, so Frequency-Conditioned Onset may be driven by domain or tokenizer artifacts like whitespace tokens or common symbols. Recompute buckets on other domains and languages, show Top-k composition, and rerun the curves.

**Questions:**

- For TunedLens, which probes are trained from scratch vs. reused from prior work, and on what data?
- In the multiple-choice setups, what exactly are the option strings the model sees (A/B/C/D characters vs. full words), and are the few-shot exemplars fixed across runs?

---

> ### Author Response · Authors · 2025-11-27
>
> We thank the reviewer for taking the time to review our work and providing actionable feedback. We have updated our paper with new experiments, specifically with causal interventions that strengthen the claims of our work. We have highlighted the major changes in the new content with blue text to reduce the cognitive load on the reviewers. We will refer to our latest uploaded version when referring to different sections, figures or lines of the paper. The responses will be split into two comments so we request the reviewer to look at both comments together.
>
> Below, we provide a response to some of the comments made by the reviewer.
>
> **WEAKNESS SECTION:**
>
> 1. We appreciate the reviewer raising the comment about the validity of TunedLens. We also spent a lot of time trying to address the issue of probe bias, which is presented in detail in Appendix section D. We summarize our experiments and results in section 5 of the main manuscript. Specifically, the question we try to answer is “whether the conclusions from the early layer decoding of internal representations reflect the information content in the activations or if it is bias introduced by TunedLens”. To answer this question, we train a custom TunedLens by masking the loss for a high-frequency token from the Top-10 bucket and artificially reducing its update frequency by a factor of 1000. We see that although the update frequency of the token goes down by 1000, its appearance in early layers goes down by a factor of 1.5. From this we conclude that the early layer representations do contain information about high-frequency tokens and this is not purely a probe bias. Based on the reviewer’s suggestion, we also perform three additional interventions on the models (described below). We hope that the probe ablations with the new interventions (see below) can further convince the reviewer of the validity of the results.
>
> 2. We appreciate the reviewers suggestion on performing causal interventions, giving us actionable feedback to improve our work. We specifically designed three interventions, one for each of the case studies in section 4. The corresponding figures for these experiments are Figure 5 and 6c.
>
>       **a)** As a summary of the first two interventions, we supplement the claims from section 4.1 and 4.2 with early-decoding experiments, showing that tokens corresponding to POS tags like DET, ADP and PUNCT more than two times more likely to match final prediction in early layers compared to content tokens in ADJ, VERB and NOUN category (lines 354-361). We see similar results for multi-token fact recall (lines 413-417).
>
>       **b)** For the third intervention, we perform activation patching to show evidence of a two-phase mechanism when a model answers a fixed-option task (MMLU/SST). We take an intermediate activation vector from SST forward pass and replace it during the MMLU forward pass (section 4.3.1). We see that when this replacement is done during the first phase of option-collection, the MMLU forward pass is able to recover the valid MMLU options (A/B/C/D). But if this replacement happens at a later layer, where the model has transitioned into the second phase of answering where it reasons between the top options choices, the final output in the MMLU forward pass is now one of the SST options (positive/negative). We believe this is a very direct causal evidence of a two-phase answering mechanism, where the model performs the easier of the two tasks in early layers of collecting valid options, and reasons between the two options in later layers.
>
>
> 3. Regarding keeping only correct predictions during fact recall, we add a new section in Appendix H in the new version paper explaining our reasons to do so. The main reason for keeping only correct options was that it allowed us to cheaply figure out if the generated answer constitutes a fact recall and the start and end point of the recalled fact. If we don’t rely on ground-truth, we are unable to recognize both these things without using a labeller. However, post reading reviews, we analyze the scenarios where the model generates incorrect answers in Appendix H by using LLM-as-a-Judge (Gemini 2.5) to figure out instances of incorrect factual recall and answer spans. We find that the qualitative results remain the same with minor differences (we refer the reviewer to Appendix H for more details). However, we want to point out that although our cleanest result holds when the model is able to do fact recall correctly, the observation that this pattern is slightly different during incorrect factual recall shows that there may be different mechanisms at play when performing correct versus incorrect factual recall. We believe this opens up avenues for future research to investigate this without taking away from current findings.
>
> We continue this response in the next comment.

---

> ### Author Response · Authors · 2025-11-27
>
> **WEAKNESS RESPONSE CONTINUED:**
>
> 4. We perform post-stability analysis in appendix E. Qualitatively we see the same results as the first crossing metric, the only difference being that the pattern appears deeper into the model. This is as per expectation since saturation is expected to occur deeper into the model. However, we would like to point to the reviewer that while we agree that first appearance is not monotonic (as also pointed out by us in footnote 3 on page 6), the information conveyed by first crossing is not spurious since we are tracking the rank of the predicted token. Thus, early progression of rank is also informative. We choose the first-crossing metric as that can present a floor for the minimum computation required to predict a token. However, confirming the findings through a post-stability metric definitely strengthens our findings and we thank the reviewer for pointing this out.
> We believe that the POS analysis done in section 4.1 shows that the results of section 3 are not solely driven by whitespace tokens or common symbols. As explained in lines 350-352, results from section 4.1 directly corroborate section 3 results.
>
> **RESPONSE TO QUESTIONS:**
>
> 1. We used off-the-shelf TunedLens models from the original authors for all experiments in the main paper. We only trained custom TunedLens for Appendix D.
> 2. The option strings for MMLU are exactly one of [A, B, C, D] and for SST it is [positive, negative]. The few shot examples across runs are the same and are chosen from a different split than the evaluation split.
>
>
> We again thank the reviewer for their comments and actionable feedback. We believe that we have provided a lot of additional evidence in response to the review that should give the reviewer additional confidence in our results. We therefore gently request the reviewer to take these results into consideration and update their scores as they find appropriate. We are also happy to engage in further discussion and answer any other questions.

---

### Official Review · Reviewer_LSpR · 2025-11-03

**Soundness:** 2
**Presentation:** 2
**Contribution:** 1
**Rating:** 2
**Confidence:** 5

**Summary:**

This paper applies TunedLens to a number of open weights LLMs and documents several observations about patterns. For example, the authors note that early layers promote frequent words while middle and later layers promote less frequent words. Later, the authors note that, for multiword tokens, the first token in the phrase doesn't emerge until a later layer, but subsequent words emerge at earlier layers. There are several other related analyses, which extend these observations to specific domains, such as fact recall and multiple choice QA.

I appreciate the author's interest in interpretability work, and I like the question they ask ("how are LLMs using their layers?") but I don't feel there is any novel contribution in this work. All of the observations the authors make boil down to some version of a well-know principle: the more context required for the prediction, the more layers required. I think this is an observation that is almost true by fiat, but which has also been documented fairly regularly over the past few years of studies that involve logit lens, tuned lens, patchscopes, and similar methods.

**Strengths:**

* The authors ask an interesting question: how do LLMs use their many layers?

**Weaknesses:**

There is no novelty or contribution here. The authors simply report a few observations that stem from a single existing method (tunedlens). In my experience working with tuned lens and similar methods, the observations described here are taken for granted, and are probably documented at least implicitly in every paper that uses these tools. Moreover, I think the basic claims made by the authors are also previously reported elsewhere, with additional mechanistic detail and insight. For example:

https://aclanthology.org/2022.emnlp-main.3/ -- documents the concept of "saturation" which is the same idea as what the present paper discusses as models "complexity aware depth use"

https://aclanthology.org/2024.naacl-long.281/ -- reports that early layers generate frequent tokens and then later refines (specific to fact recall, but does describe the mechanism for the refinement)

https://aclanthology.org/2025.naacl-long.155/ -- reports the idea that models refine as they incorporate more context in higher layers

**Questions:**

N/A

---

> ### Author Response · Authors · 2025-11-27
>
> We thank the reviewer for taking the time to comment on our work. Your comments have encouraged us to clarify the novelty of our contributions, as well as conduct new analyses, which we describe below. We respectfully ask you to take another look at the paper and reevaluate its contributions to the field in light of our comments below. We have highlighted the major changes in the new content with blue text to reduce the cognitive load on the reviewers. We will refer to our latest uploaded version when referring to different sections, figures or lines of the paper.
>
> 1. **Novelty of contributions.** The main contribution of our paper is an in-depth analysis of LLMs’ internal dynamics. We go beyond the general principle you describe (more context requires more depth) to analyze layer-by-layer predictions of LLMs across multiple case studies. We have revised the paper to highlight the parallels to and differences from past work.
>
> **What we changed:** we clarified the distinction between saturation (Geva et al, 2022) and our rank crossing metric, showing that rank crossing is a more sensitive and detailed metric (lines 498-502). We have also added a reference to Merullo et al (2024), clarifying that the fact recall dynamics described there (country -> city single token retrieval) is very different from the multi-token fact retrieval dynamic we describe (lines 508-510). Finally, we also added a reference to Lepori et al (2025): now we explicitly state that the contextualization pattern described there is broadly consistent with our “guess-then-refine” framework, although our analyses do not focus on contextualization per se (lines 514-516).
>
> 2. **Reliance on TunedLens alone.** Although we took precautions to ensure that TunedLens analyses are not an artifact of TunedLens training (Section 5) and that our rank crossing metric is stable (newly added Figure 16 in Appendix), we agree that convergent findings from multiple methods help and that causal methods are a powerful addition to probes.
>
> **What we changed:** we added an activation patching experiment (Section 4.3.1, Figure 6c) to demonstrate that the format convergence we found via TunedLens can predict which format the LLM’s final response will conform to after a causal perturbation. We also added an early exiting demonstration (Figure 5), showing that general rank trends translate into meaningful top1 prediction differences.
>
> 3. **Consistency with the reviewer’s experience.** We are happy to hear that our findings agree with your own experience using these methods. However, we expect this work to be of value to the broader AI/ML/NLP community, and so we believe it is valuable to provide an in-depth analysis of important LLM behaviors, even if they are generally consistent with interp researchers’ intuitions.
>
> **What we changed:** we highlighted the implications of our work for early exiting, using TunedLens as an early exit decoder to show differential success rate for different token types (Figure 5 and Related Work)
>
>
> Finally, we would like to highlight the contributions of this work that, according to our knowledge, were not previously reported in the literature. Please let us know if we overlooked works that showed this previously.
>
> 1. **Frequency conditioned onset (section 3):** We show that the top-ranked predictions in early LLM layers are composed primarily of high-frequency tokens, which act as statistical guesses proposed by the model early on.
>
> 2. **Multi-Token Fact Recall Dynamics (section 4.2):** We show that when recalling multi-token facts, the model requires more layers for multi-token facts than single token facts, and also that first of those multiple tokens require larger depth than the rest.
>
> 3. **Downstream Task Two-Step Processing (section 4.3):** We quantify the clear, two-phase depth usage in constrained-choice tasks. The model first uses early layers to collect all valid options into the top ranks, and then uses the later half of the model's depth exclusively for reasoning and deliberation between those choices (See Figure 5). Our newly added activation patching analysis shows that this transition phase is causally relevant.
>
> Given that other reviewers did not find our work trivial, we have hope that our contributions do have value for the community. We thank you again for your time and hope you will be able to review our updated manuscript.

---

### Author Response · Authors · 2025-11-27
**Summary of Additional Experiments and General Comments**

Based on the reviewer’s comments, we have designed three intervention experiments, one for each of the case studies in section 4. The corresponding figures for these experiments are Figure 5 and 6c, which includes a subsection 4.3.1. These experiments supplement our previous results and strengthen their claims.  We have also updated the our submission, with new content in blue colored text to minimize cognitive load for reviewers. We request the reviewers to please re-download the latest version of the submission. We provide more details below.

1. As a summary of the first two interventions, we supplement the claims from section 4.1 and 4.2 with early-decoding experiments, showing that tokens corresponding to POS tags like DET, ADP and PUNCT more than two times more likely to match final prediction in early layers compared to content tokens in ADJ, VERB and NOUN category (lines 354-361). We see similar results for multi-token fact recall (lines 413-417).

2. For the third intervention, we perform activation patching to show evidence of a two-phase mechanism when a model answers a fixed-option task (MMLU/SST). We take an intermediate activation vector from SST forward pass and replace it during the MMLU forward pass. We see that when this replacement is done during the first phase of option-collection, the MMLU forward pass is able to recover the valid MMLU options (A/B/C/D). But if this replacement happens at a later layer, where the model has transitioned into the second phased on answering where it reasons between the top options choices, the final output in the MMLU forward pass is now one of the SST options (positive/negative). We believe this is a very direct causal evidence of a two-phase answering mechanism, where the model performs the easier of the two tasks in early layers of collecting valid options, and reasons between the two options in later layers.

These new experiments show causal interventions where our findings in section 4 manifest in different experimental settings. We hope that the reviewers can incorporate these experiments and their implications in their review of our work.

---

### Author Response · Authors · 2025-12-02
**Response for Area Chair**

Dear Area Chair,

We thank you for taking the time to go through our work.

While Reviewer 3 (EivB) strongly championed the paper (Score: 8) as "systematic," "well-grounded," and "excellent" in soundness and contribution, we recognize the concerns raised by Reviewers 1 and 2 regarding novelty and causality. **In our rebuttal, we have added three major new experiments that directly address the concerns of the reviewers**. It is unfortunate that the reviewers were not able to respond to our rebuttal in time, but we believe they would have substantially raised the scores based on our responses. The paper now presents a robust, causal, and novel account of depth-wise dynamics in LLMs. We summarize our responses to the reviewers below:

**1. Addressing Novelty (Response to R1 (LSpR)):** Reviewer 1 felt the findings were merely confirming well-known principles and provided no comments on the technical content itself. We respectfully disagree with the lack of novelty claim and have clarified our contributions. Specifically, we identify three novel distinct phenomena:

* **Frequency Conditioned Onset:** Empirical proof that token frequency predicts layer-wise appearance.
* **Multi-Token Fact Recall:** A specific dynamic for how facts are retrieved across tokens.
* **Downstream Task Two-Step Processing:** A clear, quantified two-phase usage of depth in constrained tasks — now strengthened with causal manipulations.

To the best of our knowledge, these findings are not present in prior literature, especially in the papers suggested by Reviewer 1. We also note that other reviewers did not share the same concern and found the results noteworthy.


**2. Providing Causal Evidence (Response to R2 (1K5T))**: Reviewer 2’s main critique was a lack of causal validation. We have taken this very seriously and added Activation Patching experiments on MMLU and SST tasks along with early exiting experiments.

* These experiments provide direct causal evidence of the "two-phase" mechanism we proposed (early layers collecting options, later layers reasoning).

* The early exiting experiments present additional evidence that function tokens and the first token of a multi-token fact have a higher probability of being top-1 ranked tokens early into the model.



**3. Robustness of Experimental Methodology (Response to R2 & R4 (6FJo)):** We have addressed concerns regarding "survivorship bias" (analyzing only correct predictions in section 4.2) and metric brittleness:

* **New Analysis of Incorrect Predictions:** We added Appendix H, showing that our findings hold even when the model predicts incorrectly, refuting the survivorship bias concern of Reviewer 2 and 4.

* **Last-Crossing Metric:** Reviewer 2 suggested that our current metric of measuring “first crossing” is brittle. Because of this, we repeated the experiments in section 4.1 and 4.2 with the “last crossing” metric as suggested by Reviewer 2 and we see the same patterns emerge. This resolves the concern raised by reviewer 2.

* **Language Precision:** Per Reviewer 4’s suggestion, we have adjusted our terminology from "complexity-aware" to "complexity-driven" to avoid implying intent, and clarified that "refinement" refers to intermediate state changes relative to the final output. We have explicitly defined this metric in section 3.2 of the new draft.




**Conclusion:** **We strongly believe that our responses have extensively resolved the questions and concerns raised by the reviewers as summarized above.** We encourage the Area Chair to go through the responses if possible. With the addition of causal activation patching experiments, early exiting experiments and the expansion of our analysis to incorrect predictions, we have gathered the empirical evidence suggested by Reviewer 2 and 4.

We hope this clarifies the strength of our revision and warrants a re-evaluation of the lower scores.

---

### Meta-Review · Area_Chair_Zu9L · 2026-01-06

**Summary:**

This paper proposes a "Guess-then-Refine" framework to demonstrate that LLMs prioritize high-frequency token guesses in early layers before integrating context in later layers. The recommendation to reject is informed by major concerns regarding the limited findings compared to existing literature on transformer saturation (Reviewer LSpR) and the methodological reliance on non-causal probing techniques (Reviewer 1K5T), which casts doubt on whether the observed dynamics reflect true internal computation or probe artifacts.

For the benefit of this paper, we regretfully recommend rejection. Note that this is not a discouragement. The authors are encouraged to address these concerns, and we believe the paper has the potential to become a strong future submission.

**Reviewer Concerns:**

While the submission offers an analysis across varying tasks (POS tagging, fact recall, multiple choice),  addressed concerns about survivorship bia, the lack of causal validation and early-exit experiment some major concerns still remain. Specifically,


- Insight can be limited (Reviewer LSpR) The core findings are perceived to boil down to the well-known principle that "more context requires more depth," limiting the novelty. It is noted that the authors should reference related work such as "Do Language Models Use Their Depth Efficiently?" Furthermore, the authors acknowledge that the dynamic usage of layer depth across the three tasks only holds on average. A more meaningful systematic analysis would group findings by real cause rather than simply by task type.

- TMethodological Validity (Reviewer 1K5T): The reliance on TunedLens, a predictive probe, to make claims about internal computation remains a point of contention. While the authors added activation patching experiments in the rebuttal to provide causal evidence, the reviewer did not validate that these specific interventions were sufficient to overcome the limitation of the primary methodology (decoding intermediate states vs. causal computation). The concern that the probe may simply be mimicking final layer distributions rather than revealing the model's internal "decision" process is still outstanding.

**Reviewer Scores:**

Reviewer LSpR (Score: 2): Likely would have remained unchanged or increased marginally (to a 3). The critique was foundational regarding novelty ("almost true by fiat"), and while the authors clarified differences with prior art, the core philosophical objection to the contribution's significance would likely persist.

Reviewer 1K5T (Score: 2): Likely would have increased the score (to a 4 or 5). The authors directly implemented the requested causal interventions (activation patching) and stability checks. Had the reviewer participated, the reviewer would likely have acknowledged this effort. The reviewer might have retained reservations about the probe bias.

Other reviewers are likely to maintain the current score.

---

### Decision · Program_Chairs · 2026-01-26

Reject